# TEST-TIME BATCH STATISTICS CALIBRATION FOR COVARIATE SHIFT

## ABSTRACT

Deep neural networks have a clear degradation when applying to the unseen environment due to the covariate shift. Conventional approaches like domain adaptation requires the pre-collected target data for iterative training, which is impractical in real-world applications. In this paper, we propose to adapt the deep models to the novel environment during inference. An previous solution is test time normalization, which substitutes the source statistics in BN layers with the target batch statistics. However, we show that test time normalization may potentially deteriorate the discriminative structures due to the mismatch between target batch statistics and source parameters. To this end, we present a general formulation $\alpha$-BN to calibrate the batch statistics by mixing up the source and target statistics for both alleviating the domain shift and preserving the discriminative structures. Based on $\alpha$-BN, we further present a novel loss function to form a unified test time adaptation framework CORE, which performs the pairwise class correlation online optimization. Extensive experiments show that our approaches achieve the state-of-the-art performance on total twelve datasets from three topics, including model robustness to corruptions, domain generalization on image classification and semantic segmentation. Particularly, our $\alpha$-BN improves 28.4% to 43.9% on GTA5 $\rightarrow$ Cityscapes without any training, even outperforms the latest source-free domain adaptation method.

## 1 INTRODUCTION

Deep neural networks (DNNs) achieve impressive success across various applications, but heavily rely on the independent and identical distribution (i.i.d.) assumption. However, in real-world applications, the model is prone to encounter the novel instances. For examples, an automatic pilot should have robust performance under different weather conditions. Unfortunately, when applying DNNs to novel environment, the performance has a clear degradation due to the covariate shift (Ben-David et al., 2010), i.e., the test data distribution differs from the training distribution.

Domain adaptation (DA) is a promising alternative, which transfers the knowledge learned on labeled source domain to unlabeled target domain, where the data distribution is distinct (Long et al., 2015). However, domain adaptation needs the pre-collected target data, which is not applicable. Unlike DA, Domain generalization (DG) aims at training a general model from multiple source domains and generalizing to the unseen target domain. DG is more challenging since the target domain is totally unseen during training. Recently, another practical scenario named test-time adaptation (TTA) is proposed. In TTA, the target data are not pre-collected for iterative training, but used for adapting the source-trained model during inference. We show the comparison between different settings in Table 1. To make the comparison clear, we introduce two indicators. Iterative training means the model is trained on the unlabeled target data iteratively. Online training means the model parameters are updated during inference. Noticing that a recent DG work (Pandey et al., 2021) adapts the model during inference, while most previous DG methods did not. We call it as optimization-free TTA. In this paper, we focus on the practical scenarios: DG and TTA.

One of the main approaches to adapt during inference is test-time normalization (T-BN) (Nado et al., 2020). T-BN re-calculates the target batch statistics to replace the source statistics in BN layers during inference. Motivated by T-BN, Wang et al. (2021) proposed to perform test time adaptation by re-calculating the target batch statistics and updating the affine parameters in BN layers with

Table 1: The comparison between various settings. DG and TTA significantly differ from other settings since both of them get rid of the iterative training on target data.

| Setting | Source data | Target data | Iterative training | Online training |
|---|---|---|---|---|
| domain adaptation (DA) | $x_s, y_s$ | $x_t$ | ✓ | ✗ |
| source-free domain adaptation (SFDA) | - | $x_t$ | ✓ | ✗ |
| domain generalization (DG) | - | $x_t$ | ✗ | ✗ |
| fully test-time adaptation (TTA) | - | $x_t$ | ✗ | ✓ |

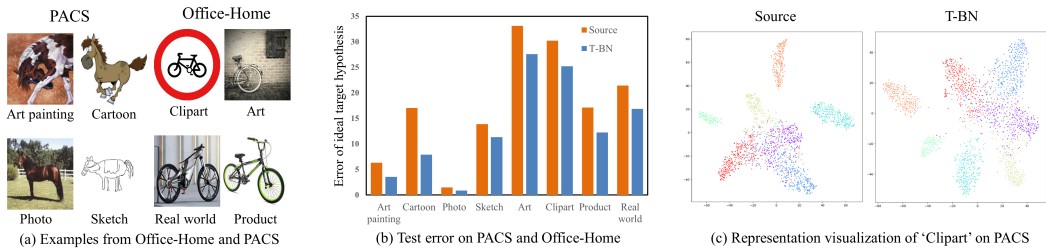

(a) Examples from Office-Home and PACS  (b) Test error on PACS and Office-Home  (c) Representation visualization of 'Clipart' on PACS

Figure 1: (Best viewed in color.) (a) Examples form two DG datasets: PACS and Office-Home. (b) Error of the ideal target hypothesis. "Art painting" indicates that the model is trained on the remaining source domains: Cartoon, Photo and Sketch, and the target representations are obtained by the source-trained model. (c) Visualization of the target representations by t-SNE (Van der Maaten & Hinton, 2008). The category cluster in T-BN shows larger variance.

entropy minimization. However, this paradigm has critical restrictions. For representation learning, the discriminative representations are crucial for recognition task. Substituting the source statistics with target statistics in BN layers will inevitably lead to a mismatch with the source-trained model parameters. This mismatch will probably perturb the original discriminative structures. Another limitation is the estimation error on target batch statistics. The source statistics in BN layers are updated in a moving average manner during training time, while the target statistics are calculated in each batch during test time. The statistics estimated in a batch introduces more errors in reflecting the domain characteristics. A preliminary empirical investigation of the mentioned restrictions are shown in Fig. 1 and Table 2.

Motivated by the hidden restrictions of T-BN, we propose a more general method named $\alpha$-BN to calibrate the batch statistics during inference. Specifically, we mix up the source and target statistics in BN layers to both alleviate the domain shift and preserve the discriminative structures. Equipped with $\alpha$-BN, common DG models can be further improved without any training. Based on $\alpha$-BN, we further propose an unified test-time adaptation framework named CORE with an online optimization, which exploits the pairwise **C**lass c**orre**lation to facilitate robust and accurate test time adaptation.

To sum up, we have following contributions:

1. We investigate two practical yet challenging transfer learning scenarios: domain generalization and test time adaptation, which release the requirement of pre-collected target domain data.

2. Motivated by the hidden restrictions of test time normalization, we present a general formulation $\alpha$-BN for both alleviating domain shifts and preserving discriminative informations. It can be seamlessly incorporated into mainstream deep neural networks to improve the generalization on unseen domains without any training.

3. Based on $\alpha$-BN, we propose a unified framework CORE for test-time adaptation. CORE optimizes the pairwise class correlation in an unsupervised online learning manner.

4. We conduct numerous experiments on total twelve datasets from three topics: robustness to corruptions, DG on image classification and DG on semantic segmentations. The empirical results show that both $\alpha$-BN and CORE achieve the state-of-the-art (SoTA) performance in their respective communities. The result on GTA5 → Cityscapes, for instance, is improved

from 28.4% to 43.9% without any training, which even outperforms the SoTA source-free DA method.

## 2 RELATED WORKS

**Domain Adaptation and Generalization** Domain adaptation (DA) enables transferring the knowledge from source domain to target domain by jointly optimizing both labeled source data and unlabeled target data. This paradigm has gained a lot of attention in the last decade and various methods are proposed, which can be roughly divided into three categories: metric learning (Long et al., 2015; Sun & Saenko, 2016; Kang et al., 2019; Li et al., 2020), adversarial training (Ganin et al., 2016; Long et al., 2018; Tsai et al., 2018) and self-training Zou et al. (2019); Liang et al. (2020); Ge et al. (2020). However, training with the large amounts source data is inefficient and impractical in many real-world applications. To address this issue, source-free DA is proposed, which adapts to the target domain with target data and the model pre-trained on source domain. Chidlovskii et al. (2016) are the first to investigate source-free DA and proposed a denoising auto-encoder for adaptation. SHOT (Liang et al., 2020) is another representative work, which proposed information maximization loss and clustering-based pseudo-labeling. However, despite getting rid of the source domain data, source-free DA also requires the pre-collected target domain data for iterative training, limiting its application scenarios.

Therefore, domain generalization (DG) and fully test-time adaptation (TTA) are proposed (Let us elaborate TTA in the next paragraph). Domain generalization aims at generalizing the model trained on a (multiple) source domain (s) to the unseen target domain directly. Recently, various DG methods are proposed including domain-invariant representation learning (Zhao et al., 2020; Matsuura & Harada, 2020), proxy tasks (Carlucci et al., 2019; Huang et al., 2020), augmentations (Volpi et al., 2018; Zhou et al., 2021), meta-learning (Li et al., 2018; Balaji et al., 2018) and so on. However, Gulrajani & Lopez-Paz (2020) provided a DG benchmark named DomainBed for fair comparison and found a well-implemented empirical risk minimization (ERM) model outperforms most DG methods. Recently, Pandey et al. (2021) proposed label-preserving target projections during inference time for DG. This work differs from most previous works, which focus on learning from source domains, while it performs optimization-free TTA during inference. Our proposed $\alpha$-BN also belongs to it, but the methods differ.

**Fully Test-time Adaptation** Fully test-time adaptation is proposed by Wang et al. (2021), which adapts the model to target domain by online training. TTA can be seen as a compromise between source-free DA and DG. Different from source-free DA, TTA does not require the pre-collected target domain data but trains on the target data in an online manner. Also, different from DG, TTA allows optimization during test, which introduces additional test time cost but usually guarantees better performance. Therefore, TTA is a really practical scenario since it gets rid of iterative training and yields better performance compared to generalizing to the new environment directly. Wang et al. (2021) proposed TENT to achieve TTA by feature modulation. Feature modulation contains two steps: test-time normalization (we will elaborate it in the next paragraph.) and affine parameters optimization by entropy minimization, which is a widely-used regularization term on DA and semi-supervised learning (Grandvalet et al., 2005). Another similar scenario is test-time training, which optimizes the networks before making a prediction during inference Sun et al. (2019). Since this setting is weaker than TTA, we mainly talk about TTA in this paper.

**Normalization and Adaptation** Batch normalization (BN) is widely-used in DNNs nowadays for stable training and fast converge. BN is originally proposed to alleviate the internal covariate shift during training a very deep neural networks Ioffe & Szegedy (2015). Recently, Schneider et al. (2020) and Nado et al. (2020) discovered that updating the batch statistics during testing improves the robustness to common corruptions. In this paper, we call it as test-time normalization. Similar to their works, Wang et al. (2021) proposed feature modulation, which also updates the batch statistics rather than freezes them. The key insight for these methods is that batch statistics are closely related to the domain characteristics (Li et al., 2016; Pan et al., 2018). Based on this finding, Jeon et al. (2021) and Zhou et al. (2021) proposed to synthesise the novel domain styles to facilitate generalization. Another kind of work focuses on instance normalization (Ulyanov et al., 2016). Similarly, Huang & Belongie (2017) found the instance-wise statistics are related to instance characteristics (i.e., image styles). Motivated by this finding, Zhou et al. (2021) proposed to generate novel in-

stances by mixing instance-level feature statistics to enhance out-of-distribution generalization. It is worth noticing that the aforementioned DG method can be summarized as an augmentation-based technique. Different from them, we propose a post-processing method to calibrate the batch statistics on target domain during test time.

## 3  UNDERSTANDING TEST-TIME NORMALIZATION

Test-time normalization (**T-BN**) re-calculates the batch statistics on target domain during inference. Since the batch statistics are closely related to the domain characteristics (Li et al., 2016; Pan et al., 2018), T-BN adapts the model to target domain explicitly. However, every coin has two sides, and T-BN is not a free lunch. During training on the source domain, the model parameters are associated with the source statistics. Therefore, substituting the source statistics with the target ones inevitably results in a mismatch with the model parameters, which leads to the degradation of discrim-

Table 2: Accuracies (%) of "Source" and "T-BN" on three DG classification benchmarks: VLCS, PACS and Office-Home.

| Method | VLCS | PACS | Office-Home |
|--------|------|------|-------------|
| Source | 77.2 | 85.3 | 66.5 |
| T-BN   | 57.9 | 83.7 | 63.9 |
| drop   | 19.3↓ | 1.6↓ | 2.6↓ |

inative structures. In short, T-BN alleviates the negative effects caused by domain shift, but perturbs the discriminative structures. We report the averaged accuracy of "Source" and "T-BN" on three DG classification benchmarks in Table 2. We observe that the accuracy of "T-BN" is consistently lower than "Source", revealing that substituting the source statistics by the estimated target batch statistics directly is not effective in generalizing to the new environment. To further understand T-BN, we begin with the following two perspectives.

**Error of ideal target hypothesis.** Based on domain adaptation theory (Ben-David et al., 2010), the domain shift can be reflected by the error of the ideal target hypothesis based on the target representations learned by source model. "target representation" means the representations are obtained on the target domain data. To obtain the ideal target hypothesis, we train a new classifier over the target representations with corresponding labels. Two methods are compared: "Source" and "T-BN". "Source" obtains the target representations by the source model directly, while "T-BN" performs test-time normalization. The error of the ideal target hypothesis is shown in Fig. 1 (b). As expected, the error of the ideal target hypothesis in "T-BN" is lower over all tasks. It is worth noticing that the only difference between them is that T-BN normalizes the BN layer inputs by the target statistics rather than the source statistics, and others remain consistent (e.g., the same network architecture and the same network parameters). Therefore, we reasonably postulate that T-BN alleviates the domain shift, which results in the lower error of the ideal target hypothesis.

**Representation visualization.** Discriminative representation learning is essential for recognition task. The discriminative representation satisfies two basic principles: intra-class tightness and inter-class separation. To qualitatively verify that how T-BN affects the learned representations, we visualize the target representations in Fig. 1 (c). The variance of each category cluster in "T-BN" is significantly larger compared to "Source", which indicates the discriminative structures are injured due to the mismatch between target statistics and source model parameters.

## 4  TEST-TIME BATCH STATISTICS CALIBRATION

Let $\{\mu_s^{(i)}, \sigma_s^{(i)}\}$ be the source statistics in $i$-th BN layers. After training on the source domain data, the BN statistics are always fixed for stable inference. However, when encountering the data with distinct distribution, the model performance usually has a clear degradation due to the covariate shift. T-BN re-calculates the target statistics $\{\mu_t^{(i)}, \sigma_t^{(i)}\}$ to replace the fixed BN statistics, which alleviates domain shift but perturbs the discriminative structures. In this paper, we present a general formulation $\alpha$-BN to generalize to new domains while preserve the discriminative structures. $\alpha$-BN considers both source and target statistics.

More specifically, $\alpha$-BN mixes the source and target statistics during test time. During test time, given an target input batch $x_t$, $\alpha$-BN re-calculates the target batch statistics $\{\mu_t^{(i)}, \sigma_t^{(i)}\}$ before

forwarding $i$-th BN layer. Then, $\alpha$-BN calibrates the batch statistics as:

$$\mu^{(i)} = \alpha\mu_s^{(i)} + (1-\alpha)\mu_t^{(i)}, \tag{1}$$

$$\sigma^{(i)} = \alpha\sigma_s^{(i)} + (1-\alpha)\sigma_t^{(i)}, \tag{2}$$

where $\alpha$ is a hyper-parameter to balance the source and target statistics. Similar formulation is also proposed by Schneider et al. (2020) for alleviating the estimated error caused by small batch size. However, we discover that even with a large batch size (e.g., 200), T-BN also yields inferior performance on the large distribution shift (e.g., Office-Home). In practice, we set $\alpha$ to 0.9 for classification tasks, and 0.7 for segmentation tasks on domain generalization benchmarks. Noticing that $\alpha$-BN is a post-processing method, and can be easily incorporated into mainstream neural networks to enhance the generalization performance on unseen domains.

Based on $\alpha$-BN, we adopt the class correlation optimization (Jin et al., 2020) for robust and accurate test-time adaptation:

$$\mathcal{L}_{CORE} = \sum_{j=1}^{C} \sum_{j' \neq j}^{C} p_{\cdot j}^{\top} p_{\cdot j'}, \tag{3}$$

where $p_{\cdot j}$ is the averaged softmax output, which indicates the probabilities that the samples in the mini-batch belongs to $j$-th class. $p_{\cdot j}^{\top} p_{\cdot j'}$ depicts the correlation between $j$-th class and $j'$-th class. Minimizing the class correlation leads to a more confident and accurate predictions like other regularization terms (e.g., entropy minimization). However, we theoretically show that CORE loss prevents the easy samples dominate the learning procedure in Appendix C, thus providing a more effective optimization-based test-time adaptation framework.

In short, we propose a unified framework named CORE for test-time adaptation based on $\alpha$-BN and class correlation. Firstly, we calibrate the batch statistics with proposed $\alpha$-BN. Then, we only optimize the affine parameters in BN layers with the loss function presented in Eq. (3). Noticing that CORE is fully unsupervised, and only updates the parameters once in each test batch. The test data will not be replayed just like the assembly line products.

## 5 EXPERIMENTS

### 5.1 DATASETS AND TASK DESIGNS

We evaluate our proposed method on 12 datasets, including three categories: robustness to corruptions, DG on image classification and DG on semantic segmentation.

**Topic 1: Robustness to corruptions.** We evaluate the robustness on CIFAR10-C, CIFAR100-C and ImageNet-C. For each dataset, 15 types of algorithmically generated corruptions with five levels of severity are included. For CIFAR10/100-C, we evaluate models on the highest severity corruptions. For ImageNet-C, the results are averaged on five levels of severity, i.e., 75 distinct corruptions.

**Topic 2: DG on image classification.** We evaluate our method on four datasets: VLCS, PACS, Office-Home and DomainNet as suggested by DomainBed (Gulrajani & Lopez-Paz, 2020).

**Topic 3: DG on semantic segmentation.** We adopts two synthetic datasets (i.e., GTA5 and SYN-THIA) and three real-world datasets (i.e., Cityscapes, BDD-100K, Mapillary).

More details about these datasets are provided in Appendix. A. Based on these datasets, we conduct numerous tasks with various base models: ResNet (He et al., 2016), WideResNet (Zagoruyko & Komodakis, 2016) with AugMix (Hendrycks* et al., 2020) and DeepLabV3 (Chen et al., 2017). The task design is shown in Table 3.

### 5.2 IMPLEMENTATION DETAILS AND BASELINES

To guarantee the reproducibility of our results, we implement the proposed methods on the widely-used benchmarks. We use RobustBench (Croce et al., 2020) for Topic 1, DomainBed (Gulrajani & Lopez-Paz, 2020) for Topic 2 and RobustNet (Choi et al., 2021) for Topic 3. Since the proposed $\alpha$-BN is training-free, we illustrate the implementation details of test-time adaptation method CORE.

Table 3: Task design.

| Category | Dataset | Model | Task description | Number of tasks |
|---|---|---|---|---|
| Model robustness | CIFAR10-C | WideResNet-28-10 | Clean CIFAR10 → Corrupted CIFAR10 (highest level) | 15 |
| | CIFAR100-C | WideResNet-40-2+AugMix | Clean CIFAR100 → Corrupted CIFAR100 (highest level) | 15 |
| | ImageNet-C | ResNet50 | Clean ImageNet → Corrupted ImageNet (total 5 levels) | 75 |
| DG on image classification | VLCS | ResNet50 | {LCS, VCS, VLS, VLC} → {V, L, C, S} | 4 |
| | PACS | ResNet50 | {ACS, PCS, PAS, PAC} → {P, A, C, S} | 4 |
| | Office-Home | ResNet50 | {CPR, APR, ACR, ACP} → {A, C, P, R} | 4 |
| | DomainNet | ResNet50 | {IPQRS, CPQRS, CIQRS, CIPRS, CIPQS, CIPQR} → {C, I, P, Q, R, S} | 6 |
| DG on semantic segmentation | see "Task description" | ResNet50 + DeepLabV3+ | {GTA5} → {SYNTHIA, Cityscapes, BDD-100K, Mapillary} | 4 |
| | see "Task description" | ResNet50 + DeepLabV3+ | {GTA5+SYNTHIA} → {Cityscapes, BDD-100K, Mapillary} | 3 |

Table 5: Test error values of different corruptions on CIFAR10-C and CIFAR100-C. The evaluation is implemented on RobustBench (Croce et al., 2020) for fair comparison and easy reproducing. We compare proposed CORE with T-BN (Schneider et al., 2020) and state-of-the-art method TENT (Wang et al., 2021). The best results are highlighted.

| Method | Dataset | Gauss. | Shot | Impulse | Defocus | Glass | Motion | Zoom | Snow | Frost | Fog | Bright | Contrast | Elastic | Pixel | JPEG | Mean |
|---|---|---|---|---|---|---|---|---|---|---|---|---|---|---|---|---|---|
| Source | CIFAR10-C | 72.3 | 65.7 | 72.9 | 46.9 | 54.3 | 34.8 | 42.0 | 25.1 | 41.3 | 26.0 | 9.3 | 46.7 | 26.6 | 58.5 | 30.3 | 43.5 |
| T-BN | CIFAR10-C | 28.1 | 26.1 | 36.3 | 12.8 | 35.3 | 14.2 | 12.1 | 17.3 | 17.4 | 15.3 | 8.4 | 12.6 | 23.8 | 19.7 | 27.3 | 20.4 |
| TENT | CIFAR10-C | 24.8 | 23.5 | 33.0 | 12.0 | 31.8 | 13.7 | 10.8 | 15.9 | 16.2 | 13.7 | 7.9 | 12.1 | 22.0 | 17.3 | 24.2 | 18.6 |
| CORE(ours) | CIFAR10-C | 22.5* | 20.3* | 29.8* | 11.0* | 29.2* | 12.3* | 10.2* | 14.4* | 14.8* | 12.4* | 7.7* | 10.6* | 20.4* | 15.3* | 21.4* | 16.8 |
| Source | CIFAR100-C | 65.7 | 60.1 | 59.1 | 32.0 | 51.0 | 33.6 | 32.3 | 41.4 | 45.2 | 51.4 | 31.6 | 55.5 | 40.3 | 59.7 | 42.4 | 46.8 |
| T-BN | CIFAR100-C | 44.3 | 44.0 | 47.3 | 32.1 | 45.8 | 32.8 | 33.0 | 38.4 | 37.9 | 45.4 | 29.8 | 36.5 | 40.6 | 36.7 | 44.1 | 39.2 |
| TENT | CIFAR100-C | 40.4 | 39.5 | 42.1 | 30.1 | 42.8 | 31.2 | 30.2 | 34.5 | 36.0 | 38.7 | 28.0 | 33.6 | 38.1 | 33.9 | 40.8 | 36.0 |
| CORE(ours) | CIFAR100-C | 39.8* | 39.3* | 41.5* | 29.5* | 41.7* | 30.6* | 29.8* | 34.2* | 34.9* | 38.6* | 27.5* | 32.6* | 37.1* | 32.7* | 40.1* | 35.3 |

We use SGD optimizer for ImageNet-C, and Adam optimizer (Kingma & Ba, 2015) for remaining tasks. We set batch size (BS) as 200, learning rate (LR) as 0.001 for CIFAR10/100, which is consistent with TENT's settings[1]. For ImageNet-C, we set BS=64, LR=2.5e-4 for fair comparison with other methods. For DG on image classification, we set BS=64, LR=1e-4 for all tasks except DomainNet, on which we reduce the LR to 1e-5.

To verify the effectiveness of proposed two methods: $\alpha$-BN and CORE, we compare them with the state-of-the-art methods on each research community. For Topic 1, test-time normalization (T-BN) (Nado et al., 2020) and test-time adaptation by entropy minimization (TENT) (Wang et al., 2021) are noteworthy approaches. For Topic 2, we incorporate $\alpha$-BN with two baselines: Empirical Risk Minimization (ERM) and CORrelation ALignment (CORAL) (Sun & Saenko, 2016). For Topic 3, we apply $\alpha$-BN to the simplest method ERM to enhance the generalization on unseen domains.

## 5.3 ROBUSTNESS TO CORRUPTIONS

The results of CIFAR10-C and CIFAR100-C are shown in Table 5. Our proposed loss consistently outperforms all existing approaches under TTA setting, which demonstrates the effectiveness of our method. To verify the performance on large-scale dataset, we evaluate our method on ImageNet-C, whose results are shown in Table 4. In the largest dataset ImageNet-C, we achieve a new state-of-the-art: 42.5% mean error over 75 tasks, which proves the superiority of our method on improving the robustness to corruption by test-time adaptation. It is worth noticing that we optimize the affine parameters in BN layers by only one iteration in an online manner. Therefore, the improvement is significant and the proposed framework CORE shows clear advantages over previous state-of-the-arts like TENT.

Table 4: Test error values of different corruptions on ImageNet-C. The best results are highlighted. The reported results are averaged over total 75 tasks. Detailed results of each task are shown in Appendix B.1.

| Method | Error (%) |
|---|---|
| Source | 59.5 |
| AugMix (Hendrycks* et al., 2020) | 51.7 |
| Norm (Schneider et al., 2020) | 49.9 |
| ANT (Rusak et al., 2020) | 50.2 |
| ANT+SIN (Geirhos et al., 2019) | 47.4 |
| TENT (Wang et al., 2021) | 44.0 |
| CORE (ours) | **42.5*** |

[1] https://github.com/DequanWang/tent

Table 6: Accuracies of state-of-the-art methods on four datasets. The evaluation is implemented on DomainBed (Gulrajani & Lopez-Paz, 2020). The best results on Optimization-Free Test-Time Adaptation (OF-TTA) and optimization-based Test-Time Adaptation (TTA) are highlighted by bold and underline, respectively.

| Method | Protocols | VLCS | PACS | Office-Home | DomainNet | Mean |
|---|---|---|---|---|---|---|
| ERM | DG | 77.5 | 85.5 | 66.5 | 40.9 | 67.6 |
| CORAL (Sun & Saenko, 2016) | DG | **78.8** | 86.2 | 68.7 | **41.5** | 68.8 |
| Mixup (Xu et al., 2020) | DG | 77.4 | 84.6 | 68.1 | 39.2 | 67.3 |
| SagNet (Nam et al., 2019) | DG | 77.8 | 86.3 | 68.1 | 40.3 | 68.1 |
| MLDG (Li et al., 2018) | DG | 77.2 | 84.9 | 66.8 | 41.2 | 67.5 |
| RSC (Huang et al., 2020) | DG | 77.1 | 85.2 | 65.5 | 38.9 | 66.7 |
| ERM† | OF-TTA | 77.2 | 85.3 | 66.5 | 40.9 | 67.5 |
| ERM†+$\alpha$-BN (ours) | OF-TTA | 77.8* | **87.9*** | 68.4* | 41.0 | 68.8 |
| CORAL† | OF-TTA | 78.2 | 86.1 | 68.4 | 41.4 | 68.5 |
| CORAL†+$\alpha$-BN (ours) | OF-TTA | 78.7* | 87.4* | **69.8*** | **41.5** | 69.4 |
| ERM†+ $\alpha$-BN + TENT (Wang et al., 2021) | TTA | 78.7* | 89.1* | 67.8* | 35.4* | 67.8 |
| CORAL†+ $\alpha$-BN + TENT (Wang et al., 2021) | TTA | 78.9* | 88.5* | 69.9* | 38.0* | 68.8 |
| ERM†+ CORE (ours) | TTA | 78.4* | 89.4* | 69.1* | 43.8* | 70.2 |
| CORAL†+ CORE (ours) | TTA | 79.2* | 88.7* | 70.4* | 43.6* | 70.5 |

## 5.4 DG ON IMAGE CLASSIFICATION

In this topic, there are two proposed methods: 'ERM+$\alpha$-BN' and 'ERM+CORE' with different evaluation protocols. Firstly, the comparison between 'ERM+$\alpha$-BN' and other DG methods is fair since $\alpha$-BN belongs to optimization-free TTA that only calibrates the batch statistics during inference without any training. Since 'ERM+CORE' belongs to optimization-based TTA, we compare it with previous state-of-the-art work TENT. Noticing that TENT is based on T-BN but T-BN has inferior performance on large distribution shift (see Table 2), we equip TENT with the proposed $\alpha$-BN for better comparison with our CORE. To emphasize the proposed $\alpha$-BN and CORE are model-agnostic, we also use CORAL (Sun & Saenko, 2016) as another baseline. The results are shown in Table 6.

$\alpha$**-BN improves the performance on unseen target domain.** Domain generalization is a really challenging task and many recent proposed methods have few improvements or even degradations compared to ERM. Equipped with our proposed $\alpha$-BN, the performance on unseen target domain is consistently improved. For instance, ERM+$\alpha$-BN reaches 68.8% and CORAL+$\alpha$-BN reaches 69.4% averaged accuracy over four DG benchmarks.

**CORE outperforms TENT with robust performance.** Equipped with $\alpha$-BN, we compare the proposed method CORE with TENT, the state-of-the-art algorithm in test-time adaptation. CORE consistently outperforms TENT over four benchmarks, which verifies the effectiveness of our method. Specifically, CORE surpass TENT by considerable margins of 2.4% and 1.7% averaged accuracy on ERM and CORAL baselines. Noticing that test-time adaptation only updates the parameters once, thus the improvement is significant. Another interesting finding is that CORE surpasses TENT by 8.4% and 5.6% accuracy on the most challenging dataset DomainNet.

## 5.5 DG ON SEMANTIC SEGMENTATION

To evaluate the universality of proposed methods, we conduct DG experiments on semantic segmentation with five large-scale datasets: GTA5, SYNTHIA, Cityscapes, BDD-100K, Mapillary. The results are shown in Table 7, and the segmentation visualization is presented in Fig. 2.

$\alpha$**-BN is embarrassingly simple but achieves state-of-the-art performance.** Previous state-of-the-art method ISW proposed instance selective whitening to remove the domain-specific styles, and reaches 40.0% mIoU over all tasks. The popular IBN-Net achieves 38.2% mIoU by utilizing both batch normalization and instance normalization. Different from previous works, we propose a post-processing method named $\alpha$-BN, which adapts the batch statistics during test time. Equipped with $\alpha$-BN, the baseline ERM reaches 40.7% mIoU over total nine tasks, yielding a new state-of-the-art on domain generalization.

Table 7: Results on semantic segmentation under domain generalization setting. We evaluate our method on two popular simulation-to-real benchmarks. '†' means the results are based on our implementation. The evaluation metrics is the mean intersection-over-union (mIoU). We compare our method with recent state-of-the-art methods: SW (Pan et al., 2019), IBN-Net (Pan et al., 2018), IterNorm (Huang et al., 2019), ISW (Choi et al., 2021) and T-BN Nado et al. (2020).

| Source | GTA5 | | | | | GTA5 + SYNTHIA | | | | Cityscapes | | | |
|---|---|---|---|---|---|---|---|---|---|---|---|---|---|
| Target | SYNTHIA | Cityscapes | BDD-100K | Mapillary | mean | Cityscapes | BDD-100K | Mapillary | mean | BDD-100K | Mapillary | mean | MEAN |
| ERM | 26.2 | 29.0 | 25.1 | 28.2 | 27.1 | 35.5 | 25.1 | 31.9 | 30.8 | 45.0 | 51.7 | 48.4 | 33.1 |
| SW | 27.6 | 29.9 | 27.5 | 29.7 | 28.7 | - | - | - | - | 48.5 | 55.8 | 52.2 | - |
| IBN-Net | 27.9 | 33.9 | 32.3 | 37.8 | 33.0 | 35.6 | 32.2 | 38.1 | 35.3 | 48.6 | 57.0 | 52.8 | 38.2 |
| IterNorm | 27.1 | 31.8 | 32.7 | 33.9 | 31.4 | - | - | - | - | 49.2 | 56.3 | 52.8 | - |
| ISW | 28.3 | 36.6 | **35.2** | 40.3 | 35.1 | 37.7 | 34.1 | 38.5 | 36.8 | 50.7 | 58.6 | 54.7 | 40.0 |
| ERM† | 26.7 | 28.4 | 24.3 | 27.9 | 26.8 | 36.2 | 24.3 | 31.5 | 30.7 | 45.9 | 52.5 | 49.2 | 33.1 |
| T-BN | 27.2 | 41.0 | 32.5 | 38.1 | 34.7 | 44.0 | 34.3 | 39.3 | 39.2 | 47.6 | 48.2 | 47.9 | 39.1 |
| α-BN(ours) | **28.7** | **43.9** | 33.4 | 38.2 | **36.1** | **44.8** | **35.0** | **40.3** | **40.0** | 49.3 | 52.6 | 51.0 | **40.7** |

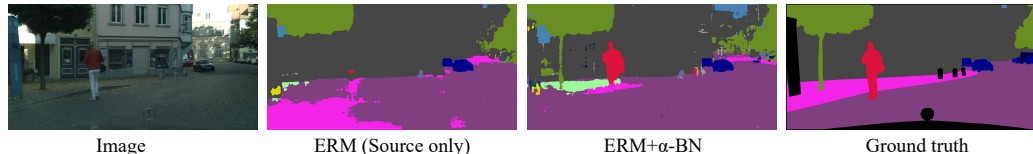

| Image | ERM (Source only) | ERM+α-BN | Ground truth |

Figure 2: (Best viewed in color.) Qualitative results on GTA5 → Cityscapes. Equipped with our α-BN, the source model yields better and cleaner segmentation map on target domain with little additional test time (e.g., about 16ms on each image). Specifically, without any training, α-BN re-discovers the missing class "person", which is colored in red. More qualitative results are shown in Appendix D.5.

**α-BN even outperforms the source-free DA method without any training.** Source-free DA (Liang et al., 2020) becomes a popular topic in DA community recently. For semantic segmentation, SFDA (Liu et al., 2021) shares similar backbone as ours, and reaches 43.2% mIoU with the 34.1% mIoU ERM baseline on GTA5 → Cityscapes. With weaker ERM baseline (28.3% mIoU), the proposed α-BN even achieves better performance (43.9% mIoU).

# 6 ANALYSIS

**α-BN introduce little additional inference time.** To evaluate the efficiency of proposed α-BN, we calculate the wall-clock time on GTA5 → Cityscapes with the same running environments for five times. The averaged wall-clock times for vanilla inference and α-BN are 72.84s and 80.94s, respectively. With little additional inference time cost (i.e., additional 0.0158s on each image), the proposed α-BN brings 15.5% mIoU improvement on GTA5 → Cityscapes.

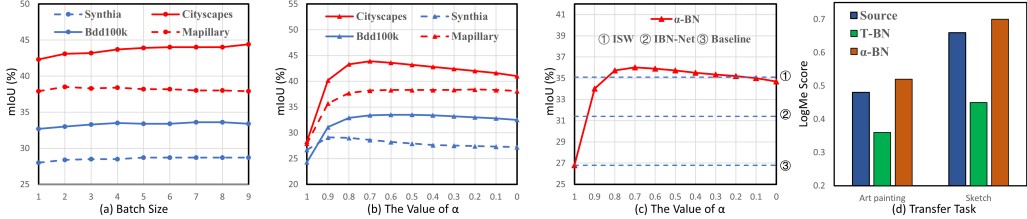

Figure 3: (Best viewed in color.) Analysis of batch size, parameter sensitivity and LogME score.

**Test-time batch size.** Fig 3 (a) shows the results under different batch size on four DG segmentation tasks. We observe that the performance is robust to test-time batch size (BS). For classification task, the observation is similar to (Schneider et al., 2020): the performance begins robust when BS ≥ 64.

**The hyper-parameter α is robust to task.** There are only one hyper-parameter in our proposed α-BN. To evaluate the parameter sensitivity, we illustrate the results on four segmentation tasks with different values of α in Fig. 3 (b). We observe that α = 0.7 is a great choice for all DG segmentation tasks, and we also observe that α = 0.9 is robust for DG classification tasks. We also present the comparison with other methods in Fig. 3 (c). The mIoU in Fig. 3 (c) is averaged on four

tasks $\{$GTA5$\} \to \{$SYNTHIA, Cityscapes, BDD-100K, Mapillary$\}$. We observe that $\alpha$-BN ($\alpha = 0.7$) outperforms all alternatives. It is worth noticing that $\alpha$-BN is training-free, and only introduces little additional inference time based on the ERM mdoel, while other methods like ISW has additional loss functions and augmentations.

**$\alpha$-BN gains more transferable representations.** The LogME score is a practical assessment for the transferability of representations (You et al., 2021). Fig. 3 (d) shows the LogME score on tasks $\{$P,C,S$\} \to \{$A$\}$, $\{$P,A,C$\} \to \{$S$\}$ with the obtained representations of Source, T-BN and the proposed $\alpha$-BN. We observe that the LogME score on $\alpha$-BN representations is larger than the LogME score on Source and T-BN representations. The finding implies that $\alpha$-BN gains more transferable representations.

**$\alpha$-BN reaches statistical significance.** To verify the statistical significance of $\alpha$-BN, we conduct the McNemars Test (Dietterich, 1998) to show the significant improvement brought by the proposed $\alpha$-BN contrast to the baseline ERM. The $p$-values for four tasks (the order of tasks is the same as Table B.4) on Office-Home are $3.4\times10^{-5}$, $2.4\times10^{-13}$, $0.0012$ and $0.0016$, respectively. As is widely acknowledged, we set the significance threshold as 0.05. Therefore, the significance test indicates our $\alpha$-BN brings significant improvement to the common baselines like ERM. We also conduct statistical significance test on other benchmarks. The result with '*' indicates that it is statistical significant.

# 7 DISCUSSION

**Why CORE ($\alpha$-BN+CORE loss) outperforms TENT (T-BN+Entropy loss)?** Firstly, T-BN leads to a mismatch between target statistics and source-learned parameters, while $\alpha$-BN balances the covariate shift alleviation and discriminative structure preservation. More essentially, $\alpha$-BN provides a better initialization of BN parameters. However, we show that a better initialization is important for test-time adaptation (see Table 2, 8 and 9). Since the supervision signals come from the model predictions, a better initialization leads to a better supervision signals in the online unsupervised learning setting. In Appendix C, we provide a theoretical insight on why CORE loss outperforms Entropy loss. Compare to Entropy loss, CORE loss has much smaller gradient on the easy samples, preventing easy samples dominate the learning procedure. Extensive empirical results further demonstrate it (see Table 4, 5, 6, 8 and 9).

**$\alpha$-BN v.s. T-BN** From Figure 3 (b,c), $\alpha$-BN has limited improvement compared to T-BN, which is inconsistent with the finding provided in Table 2. In fact, they are not contradictory. Previous work TENT has shown that T-BN performs well on the synthetic-to-real adaptation (Figure 3 (b,c)). However, in this paper, we first find that T-BN is not suitable for the nature distribution shift of real-world datasets (Table 2). We think the latter one is more important since distribution shift is a common phenomenon in real world. We have also provide the real-to-real adaptation results on semantic segmentation in Table 7. We observe that $\alpha$-BN outperforms T-BN by a margin (3.1 mIoU).

**How to determine the value of $\alpha$ for a new dataset?** The value of $\alpha$ plays an essential role in $\alpha$-BN and CORE. In this paper, we perform grid search, and find an empirical rule: $\alpha = 0.9$ for classification and $\alpha = 0.7$ for semantic segmentation (see Figure 3 and 5). Actually, this setting is not an optimal but an acceptable choice for each task, and always outperforms the original T-BN. We think developing the learnable $\alpha$-BN with an automatically adjusted $\alpha$ is an inspiring direction for future work.

# 8 CONCLUSION

We present a general formulation named $\alpha$-BN to generalize deep models to the novel environments. By considering both source and target statistics, $\alpha$-BN alleviates the covariate shift but preserves discriminative structures. Based on $\alpha$-BN, we further propose an unified test-time adaptation approach CORE, which provides a robust optimization by investigating the pair-wise correlations. Different from conventional domain adaptation approaches, our $\alpha$-BN and CORE are online algorithms, which adapts to the target domain during inference. Comprehensive experiments on twelve datasets from three research topics validate the effectiveness and efficiency of our methods.

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

## A DATASETS DESCRIPTION

We evaluate our proposed method on 12 datasets from three communities: robustness to corruptions (CIFAR10/100-C, ImageNet-C), DG on image classification (VLCS, PACS, Office-Home, Terra Incognita, DomainNet) and DG on semantic segmentation (GTA5, SYNTHIA, Cityscapes, BDD-100K, Mapillary). Some examples are illustrated in Fig. 4. Here are the detailed descriptions:

**CIFAR10/100-C** (Krizhevsky et al., 2009) has 50000 training samples and 10000 test samples with 10/100 classes. The corruptions include gaussian noise, shot noise, impulse noise, defocus blur,

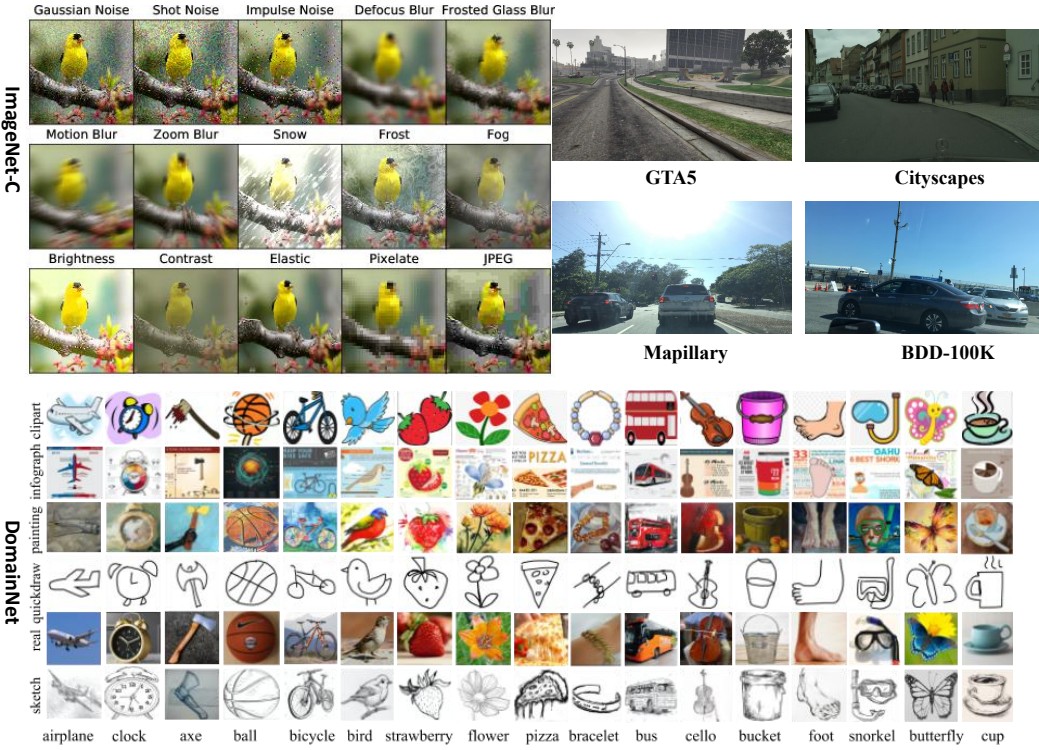

Figure 4: (Best viewed in color.) Some examples collected from the datasets: ImageNet-C, DomainNet, GTA5, Cityscapes, BDD-100K and Mapillary. The ImageNet-C figure is borrow from (Hendrycks & Dietterich, 2019) and the DomainNet figure is borrow from (Peng et al., 2019).

frosted glass blur, motion blur, zoom blur, snow, frost, fog, brightness, contrast, elastic, pixelate and JPEG. (Hendrycks & Dietterich, 2019)

**ImageNet-C** (Russakovsky et al., 2015; Hendrycks & Dietterich, 2019) has 1.2 million training samples and 50000 validation samples with 1000 classes. The corruption categories are the same as CIFAR10/100-C.

**VLCS** (Fang et al., 2013) contains 10,729 examples from four domains: Caltech101 (C), LabelMe (L), SUN09 (S), VOC2007 (V), with 5 classes.

**PACS** (Li et al., 2017) contains 9,991 examples from four domains: Art (A), Cartoons (C), Photos (P), Sketches (S), with 7 classes.

**Office-Home** (Venkateswara et al., 2017) contains 15,588 examples from four domains: Art (A), Clipart (C), Product (P), Real (R), with 65 classes.

**DomainNet** (Peng et al., 2019) is the largest dataset on DG community, and contains 586,575 examples from six domains: clipart (C), infograph (I), painting (P), quickdraw (Q), real (R), sketch (S), with 345 classes.

**GTA5** (Richter et al., 2016) is a large-scale synthetic dataset, which contains 24,966 high-resolution images collected from the game, Grand Theft Auto V (GTA5), and the corresponding ground-truth segmentation maps are generated by computer graphics. It shares 19 classes with Cityscapes.

**SYNTHIA** (Ros et al., 2016) is also a synthetic dataset rendered from a virtual city and comes with pixel-level segmentation annotations. We adopt the subset of it, which is called SYNTHIA-RAND-CITYSCAPES. This subset contains 9400 images and shares 16 common classes with Cityscapes dataset.

**Cityscapes** (Cordts et al., 2016) is the widely-used real-world dataset, which contains 3,450 high-resolution (i.e., 2048×1024) urban driving scene images.

**BDD-100K** (Yu et al., 2020) is a recent real-world dataset containing 8000 urban scene images (7000 images for training and 1000 images for validation) with resolution of 1280×720.

**Mapillary** (Neuhold et al., 2017) is another recent real-world dataset containing 25,000 street-view images with a minimum resolution of 1920×1080.

## B  FULL RESULTS

We implement our methods with Pytorch on a single NVIDIA RTX 3090 GPU. All results in this section are obtained by our implementations.

### B.1  IMAGENET-C

All results are statistical significant compared to baseline.

| Corruptions | Level 1 | Level 2 | Level 3 | Level 4 | Level 5 | Mean |
|---|---|---|---|---|---|---|
| Guass. | 32.33 | 37.13 | 45.00 | 55.15 | 69.06 | 47.73 |
| Shot | 32.83 | 38.09 | 45.08 | 57.57 | 67.19 | 48.15 |
| Impulse | 36.05 | 41.44 | 45.97 | 56.16 | 68.06 | 49.53 |
| Defocus | 34.13 | 39.33 | 50.59 | 60.67 | 70.60 | 51.06 |
| Glass | 33.26 | 39.22 | 52.83 | 58.89 | 70.23 | 50.89 |
| Motion | 30.32 | 34.06 | 41.03 | 49.81 | 57.08 | 42.46 |
| Zoom | 34.83 | 39.12 | 41.93 | 45.66 | 49.74 | 42.26 |
| Snow | 34.71 | 43.64 | 43.36 | 50.06 | 51.38 | 44.36 |
| Frost | 34.23 | 43.70 | 51.28 | 52.38 | 57.61 | 47.84 |
| Fog | 30.23 | 31.63 | 33.88 | 36.05 | 42.12 | 34.78 |
| Bright | 26.48 | 27.44 | 28.73 | 30.60 | 33.01 | 29.25 |
| Contrast | 28.73 | 30.42 | 33.35 | 43.01 | 66.79 | 40.46 |
| Elastic | 30.88 | 44.32 | 31.03 | 34.28 | 44.17 | 36.94 |
| Pixel | 28.82 | 29.70 | 32.93 | 37.88 | 41.14 | 34.09 |
| JPEG | 30.37 | 32.81 | 34.49 | 39.68 | 47.22 | 36.91 |
| Mean | 31.88 | 36.80 | 40.77 | 47.19 | 55.69 | **42.47** |

### B.2  VLCS

| Algorithm | C | L | S | V | Avg. |
|---|---|---|---|---|---|
| ERM | 97.5 | 63.4 | 73.8 | 74.0 | 77.2 |
| ERM+$\alpha$-BN | 96.3* | 67.7* | 73.5* | 73.5* | 77.8 |
| ERM+TENT | 97.3* | 69.7* | 75.1* | 72.8* | 78.7 |
| ERM+CORE | 97.0* | 67.7* | 76.6* | 72.4* | 78.4 |
| CORAL | 97.9 | 66.1 | 73.2 | 75.4 | 78.2 |
| CORAL+$\alpha$-BN | 97.0* | 69.3* | 74.4* | 73.9* | 78.7 |
| CORAL+TENT | 97.4* | 69.1* | 74.5* | 74.6* | 78.9 |
| CORAL+CORE | 97.3* | 69.1* | 75.7* | 74.7* | 79.2 |

### B.3  PACS

| Algorithm | A | C | P | S | Avg. |
|---|---|---|---|---|---|
| ERM | 84.7 | 80.2 | 96.9 | 79.2 | 85.3 |
| ERM+$\alpha$-BN | 88.2* | 83.5* | 97.5* | 82.4* | 87.9 |
| ERM+TENT | 90.4* | 83.5* | 97.7* | 84.8* | 89.1 |
| ERM+CORE | 90.4* | 83.8* | 97.7* | 85.8* | 89.4 |
| CORAL | 88.2 | 80.0 | 97.3 | 78.8 | 86.1 |
| CORAL+$\alpha$-BN | 89.0* | 82.5* | 97.7* | 80.2* | 87.4 |
| CORAL+TENT | 90.4* | 82.9* | 97.9* | 82.9* | 88.5 |
| CORAL+CORE | 90.5* | 82.8* | 97.9* | 83.6* | 88.7 |

## B.4 OFFICE-HOME

| Algorithm | A | C | P | R | Avg. |
|---|---|---|---|---|---|
| ERM | 61.2 | 52.7 | 75.8 | 76.3 | 66.5 |
| ERM+$\alpha$-BN | 63.2* | 56.1* | 76.9* | 77.3* | 68.4 |
| ERM+TENT | 62.6* | 54.4* | 77.2* | 77.1* | 67.8 |
| ERM+CORE | 63.6* | 57.9* | 77.3* | 77.6* | 69.1 |
| CORAL | 64.3 | 54.7 | 76.6 | 78.0 | 68.4 |
| CORAL+$\alpha$-BN | 64.6* | 58.4* | 77.2* | 78.9* | 69.8 |
| CORAL+TENT | 64.7* | 58.5* | 77.8* | 78.7* | 69.9 |
| CORAL+CORE | 65.3* | 60.1* | 77.5* | 78.7* | 70.4 |

## B.5 DOMAINNET

| Algorithm | cli | info | paint | quick | real | sketch | Avg. |
|---|---|---|---|---|---|---|---|
| ERM | 57.9 | 19.1 | 46.7 | 12.4 | 59.6 | 49.8 | 40.9 |
| ERM+$\alpha$-BN | 57.9 | 19.1 | 46.7 | 12.4 | 60.1 | 49.7 | 41.0 |
| ERM+TENT | 59.1* | 11.2* | 30.5* | 1.8* | 60.2* | 49.4* | 35.4 |
| ERM+CORE | 59.8* | 20.7* | 50.1* | 17.9* | 61.5* | 53.0* | 43.8 |
| CORAL | 59.2 | 19.8 | 47.3 | 13.1 | 59.0 | 50.2 | 41.4 |
| CORAL+$\alpha$-BN | 59.2 | 19.8 | 47.3 | 13.1 | 59.1 | 50.2 | 41.5 |
| CORAL+TENT | 59.0* | 11.7* | 44.1* | 3.2* | 58.5* | 51.2* | 38.0 |
| CORAL+CORE | 60.8* | 21.3* | 48.1* | 17.1* | 61.5* | 53.0* | 43.6 |

## B.6 DG ON SEMANTIC SEGMENTATION

| Method | road | side. | build. | wall* | fence* | pole* | light | sign | vege. | terr. | sky | pers. | rider | car | truck | bus | train | motor | bike | mIoU | gain |
|---|---|---|---|---|---|---|---|---|---|---|---|---|---|---|---|---|---|---|---|---|---|
| | | | | | | | | GTA5 → SYNTHIA | | | | | | | | | | | | | |
| ERM | 45.2 | 38.4 | 75.5 | 4.5 | 3.6 | 21.6 | 13.4 | 9.3 | 59.4 | 0.0 | 88.8 | 56.0 | 6.1 | 51.2 | 0.0 | 12.8 | 0.0 | 14.8 | 7.6 | 26.7 | - |
| $\alpha$-BN | 51.2 | 29.5 | 80.7 | 7.6 | 10.1 | 28.8 | 15.4 | 11.2 | 61.0 | 0.0 | 89.5 | 52.4 | 7.9 | 51.5 | 0.0 | 25.5 | 0.0 | 14.8 | 7.3 | **28.7** | **+2.0** |
| | | | | | | | | GTA5 → Cityscapes | | | | | | | | | | | | | |
| ERM | 39.7 | 23.8 | 52.9 | 16.0 | 17.5 | 23.8 | 30.7 | 14.5 | 81.1 | 27.2 | 39.8 | 58.6 | 6.4 | 57.2 | 18.5 | 14.0 | 1.0 | 7.3 | 9.3 | 28.4 | - |
| $\alpha$-BN | 87.0 | 38.7 | 83.3 | 30.3 | 27.4 | 35.1 | 36.4 | 24.6 | 82.8 | 30.0 | 77.4 | 65.6 | 23.9 | 85.9 | 30.8 | 27.7 | 5.2 | 16.8 | 26.3 | **43.9** | **+15.5** |
| | | | | | | | | GTA5 → BDD-100K | | | | | | | | | | | | | |
| ERM | 43.7 | 21.7 | 32.8 | 3.5 | 21.0 | 27.1 | 29.2 | 17.3 | 58.4 | 21.0 | 31.0 | 42.6 | 4.3 | 66.9 | 12.9 | 4.8 | 0.0 | 14.1 | 10.2 | 24.3 | - |
| $\alpha$-BN | 76.4 | 27.3 | 60.0 | 8.9 | 24.5 | 33.5 | 35.2 | 22.3 | 63.9 | 20.4 | 68.8 | 42.1 | 7.6 | 76.9 | 17.4 | 11.8 | 0.0 | 12.9 | 24.3 | **33.4** | **+9.1** |
| | | | | | | | | GTA5 → Mapillary | | | | | | | | | | | | | |
| ERM | 45.3 | 24.4 | 32.7 | 6.5 | 17.0 | 28.0 | 35.4 | 8.1 | 66.0 | 24.6 | 40.4 | 53.0 | 4.8 | 72.4 | 23.8 | 5.1 | 10.1 | 12.5 | 20.2 | 27.9 | - |
| $\alpha$-BN | 75.7 | 38.3 | 48.0 | 14.6 | 22.6 | 36.1 | 38.8 | 36.2 | 71.5 | 25.4 | 61.2 | 50.8 | 16.5 | 79.2 | 30.4 | 19.7 | 9.9 | 25.4 | 24.8 | **38.2** | **+10.3** |
| | | | | | | | | GTA5 + SYNTHIA → Cityscapes | | | | | | | | | | | | | |
| ERM | 74.5 | 36.7 | 66.5 | 11.5 | 3.0 | 31.4 | 35.8 | 21.4 | 84.7 | 10.8 | 73.2 | 66.2 | 12.0 | 84.6 | 15.7 | 25.7 | 0.0 | 12.0 | 21.6 | 36.2 | - |
| $\alpha$-BN | 85.4 | 44.1 | 84.0 | 27.8 | 11.4 | 41.2 | 41.9 | 32.1 | 85.7 | 27.7 | 87.3 | 65.0 | 19.2 | 87.0 | 22.9 | 31.1 | 0.1 | 22.4 | 34.2 | **44.8** | **+12.6** |
| | | | | | | | | GTA5 + SYNTHIA → BDD-100K | | | | | | | | | | | | | |
| ERM | 40.7 | 27.1 | 33.3 | 1.9 | 6.9 | 28.8 | 38.3 | 19.4 | 63.6 | 8.4 | 44.6 | 51.8 | 13.4 | 61.4 | 0.9 | 4.0 | 0.0 | 5.5 | 11.6 | 24.3 | - |
| $\alpha$-BN | 74.9 | 30.4 | 60.3 | 5.8 | 18.4 | 36.8 | 40.4 | 32.4 | 71.3 | 21.6 | 77.8 | 39.9 | 13.0 | 76.5 | 7.6 | 14.7 | 0.0 | 18.2 | 24.1 | **35.0** | **+10.7** |
| | | | | | | | | GTA5 + SYNTHIA → Mapillary | | | | | | | | | | | | | |
| ERM | 61.0 | 36.7 | 33.8 | 9.3 | 7.7 | 29.8 | 39.4 | 11.6 | 78.3 | 38.0 | 64.6 | 59.7 | 5.9 | 78.5 | 6.6 | 4.9 | 0.1 | 9.2 | 22.9 | 31.5 | - |
| $\alpha$-BN | 74.4 | 39.1 | 52.1 | 17.1 | 15.6 | 40.4 | 46.6 | 44.4 | 79.2 | 42.4 | 72.6 | 48.3 | 14.1 | 77.5 | 25.3 | 17.5 | 1.0 | 24.7 | 33.6 | **40.3** | **+8.8** |

## C  THEORETICAL INSIGHT ON CORE LOSS

In this paper, we adopt the CORE loss with class correlation optimization rather than the Entropy loss suggested by TENT (Wang et al., 2021). From Table 4, 5 and 6, we observe that CORE loss outperforms Entropy loss on a wide range of benchmarks. To better understand CORE loss, we provide a theoretical insight on why CORE loss outperforms Entropy loss. Firstly, the CORE loss in Eq. (3) is equal to:

$$\mathcal{L}_{CORE} = 1 - \sum_{j}^{C} p_j p_j, \tag{4}$$

where $p_j$ denotes the probability of class $j$. The entropy minimization loss is defined as:

$$\mathcal{L}_{Entropy} = -\sum_{j}^{C} p_j \log(p_j). \tag{5}$$

Consider a binary classification problem, we can obtain the values and the gradients with respect to Entropy loss and CORE loss:

$$\mathcal{L}_{Entropy} = -p \log p - (1-p) \log(1-p), \tag{6}$$

$$\frac{d\mathcal{L}_{Entropy}}{dp} = \log(1-p) - \log p. \tag{7}$$

$$\mathcal{L}_{CORE} = 1 - p^2 - (1-p)^2, \tag{8}$$

$$\frac{d\mathcal{L}_{CORE}}{dp} = 2 - 4p. \tag{9}$$

From Eq. (7) and Eq. (9), the absolute value of Entropy loss gradient on the easy sample ($p > 0.9$ or $p < 0.1$) is much larger than the absolute value of CORE loss gradient. In other words, Entropy loss pays too much attention to the easy samples.

## D  MORE EXPERIMENTAL RESULT AND ANALYSIS

### D.1  ABLATION STUDY

CORE contains two components: $\alpha$-BN and Class Correlation Optimization (COO), while TENT also contains two components: T-BN and Entropy Minimization (EM). In Table 6, we compare $\alpha$-BN + COO with $\alpha$-BN + EM, and show that COO significantly outperforms EM. In this section, we further evaluate remaining two variants: T-BN + COO and T-BN + EM (Original TENT). From Table 8, we observe that T-BN works badly on the nature distribution shift of real-world datasets. Equipped with T-BN, COO achieves similar performance with EM, and both of them are worse than the ERM baseline. This ablation study indicates that $\alpha$-BN is really important for adapting the model to a novel target domain.

Table 8: Results of four optimization-based TTA methods on VLCS, PACS and Office-Home.

| Dataset | VLCS | | | | | PACS | | | | | Office-Home | | | | |
|---|---|---|---|---|---|---|---|---|---|---|---|---|---|---|---|
| Algorithm | C | L | S | V | Avg. | A | C | P | R | Avg. | A | C | P | R | Avg. |
| ERM | 97.5 | 63.4 | 73.8 | 74.0 | 77.2 | 84.7 | 80.2 | 96.9 | 79.2 | 85.3 | 61.2 | 52.7 | 75.8 | 76.3 | 66.5 |
| T-BN | 66.9 | 49.3 | 53.2 | 62.1 | 57.9 | 87.4 | 81.3 | 95.6 | 74.6 | 84.7 | 61.2 | 50.4 | 72.0 | 76.1 | 64.9 |
| T-BN + EM | 68.7 | 50.9 | 52.5 | 63.6 | 58.9 | 87.8 | 81.5 | 95.6 | 75.6 | 85.1 | 61.8 | 51.7 | 73.0 | 76.4 | 65.7 |
| T-BN + COO | 68.5 | 50.3 | 54.6 | 62.4 | 59.0 | 87.8 | 81.8 | 95.6 | 75.2 | 85.1 | 61.8 | 51.8 | 73.0 | 76.6 | 65.8 |
| CORE (ours) | 97.0 | 67.7 | 76.6 | 72.4 | 78.4 | 90.4 | 83.8 | 97.9 | 85.8 | 89.4 | 63.6 | 57.9 | 77.3 | 77.6 | 69.1 |

## D.2 Result on GTA5 → Cityscapes with Another Backbone

ResNet101+DeepLabV2 is the popular backbone adopted by many domain adaptation methods. To better evaluate our $\alpha$-BN, we report the performance with this backbone on the most popular transfer task: GTA5 → Cityscapes. The mIoU of ERM baseline is 35.7%. $\alpha$-BN achieves 44.7% mIoU, which is also competitive state-of-the-art DG method FSDR (Huang et al., 2021) with a 44.8% mIoU.

Table 9: Results of four optimization-based TTA methods on GTA5 → Cityscapes.

| Method | road | side. | build. | wall* | fence* | pole* | light | sign | vege. | terr. | sky | pers. | rider | car | truck | bus | train | motor | bike | **mIoU** | **gain** |
|---|---|---|---|---|---|---|---|---|---|---|---|---|---|---|---|---|---|---|---|---|---|
| ERM | 39.7 | 23.8 | 52.9 | 16.0 | 17.5 | 23.8 | 30.7 | 14.5 | 81.1 | 27.2 | 39.8 | 58.6 | 6.4 | 57.2 | 18.5 | 14.0 | 1.0 | 7.3 | 9.3 | 28.4 | - |
| T-BN | 85.4 | 41.9 | 81.5 | 26.2 | 23.0 | 33.3 | 35.0 | 20.7 | 74.9 | 20.7 | 75.6 | 62.2 | 26.3 | 83.7 | 26.4 | 25.8 | 1.4 | 15.0 | 20.6 | 41.0 | +12.6 |
| T-BN+EM (TENT) | 86.5 | 43.9 | 80.6 | 24.7 | 21.3 | 32.8 | 33.1 | 19.5 | 75.4 | 22.5 | 74.8 | 60.4 | 23.6 | 82.6 | 26.8 | 22.4 | 1.5 | 16.0 | 18.4 | 40.4 | +12.0 |
| T-BN+COO | 86.4 | 44.0 | 80.8 | 24.4 | 22.5 | 35.0 | 37.4 | 23.9 | 74.4 | 21.2 | 74.4 | 60.9 | 25.5 | 82.6 | 25.5 | 23.1 | 1.6 | 17.4 | 19.7 | 41.1 | +12.7 |
| $\alpha$-BN | 87.0 | 38.7 | 83.3 | 30.3 | 27.4 | 35.1 | 36.4 | 24.6 | 82.8 | 30.0 | 77.4 | 65.6 | 23.9 | 85.9 | 30.8 | 27.7 | 5.2 | 16.8 | 26.3 | 43.9 | +15.5 |
| $\alpha$-BN+EM | 88.5 | 40.7 | 82.8 | 29.4 | 21.9 | 33.9 | 29.7 | 19.7 | 83.2 | 30.2 | 79.0 | 65.5 | 28.1 | 85.6 | 30.0 | 27.3 | 1.7 | 18.0 | 22.8 | 43.1 | +14.7 |
| $\alpha$-BN+COO (CORE) | 88.8 | 43.4 | 83.5 | 28.6 | 27.0 | 40.5 | 42.4 | 33.1 | 82.7 | 27.6 | 77.0 | 65.6 | 29.6 | 85.6 | 25.5 | 28.0 | 2.7 | 17.7 | 26.5 | 45.0 | +16.6 |

## D.3 Result of Optimization-based TTA methods on GTA5 → Cityscapes

We experiment the optimization-based TTA methods TENT, CORE and their variants on the semantic segmentation task GTA5 → Cityscapes. We set batch size as 3 and learning rate as 1e-5 with Adam optimization. The results are shown in Table 9. We observe that COO outperforms EM with both T-BN and $\alpha$-BN. $\alpha$-BN+COO achieves the best performance, demonstrating effectiveness of each component of CORE. However, the single $\alpha$-BN has already achieves 43.9% mIoU, COO appears to have limited improvement, and EM even leads to negative transfer. Recent advances in domain adaptive semantic segmentation also revealed that the universal unsupervised loss is not suitable for semantic segmentation, since this task has many intrinsic challenges like class-imbalance, boundary confusion and so on. Inspired by that self-training has become the dominant method in domain adaptive semantic segmentation, we think a well-designed self-training loss equipped with our $\alpha$-BN is a hopeful direction for test-time adaptive semantic segmentation.

## D.4 Parameter Sensitivity of Hyper-parameter $\alpha$

The value of hyper-parameter $\alpha$ plays an essential role in $\alpha$-BN and CORE. In this paper, we determine the value of $\alpha$ through grid search: from 0 to 1 with an interval of 0.1. The result of gird search on semantic segmentation is illustrated in Figure. 3. For image classification, we find the performance usually drops when $\alpha$ is decreasing from 0.9 to 0. Actually, $\alpha = 0.9$ is not the optimal choice for each task, but an acceptable choice. We further provide the result on image classification in Figure. 5. We think developing the learnable $\alpha$-BN with an automatically adjusted $\alpha$ is an inspiring direction for future work.

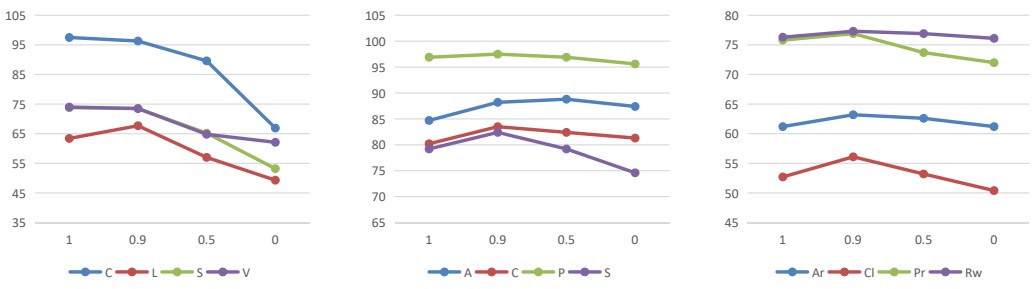

Figure 5: (Best viewed in color.) The results of grid search on classification datasets VLCS, PACS and Office-Home. The horizontal coordinate represents the value of $\alpha$ and the vertical coordinate represents the classification accuracy.

## D.5 Qualitative Results on GTA5 → Cityscapes

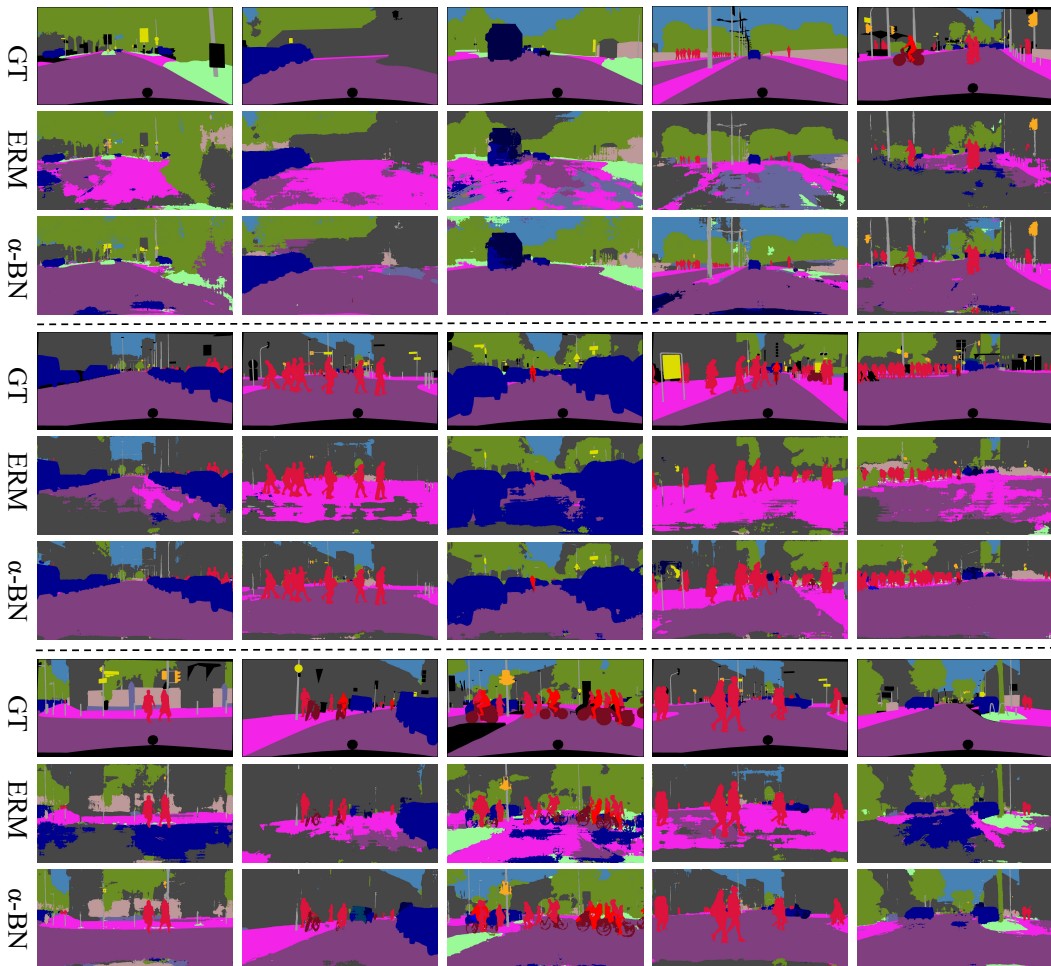

Figure 6: (Best viewed in color.) Qualitative results on GTA5 → Cityscapes. "GT" means ground truth. Equipped with our $\alpha$-BN, the segmentation performance on new environment is significantly improved with little additional test time cost (e.g., about 16ms on each image).

