# OpenReview forum: "Test-time Batch Statistics Calibration for Covariate Shift"
_ICLR.cc/2022/Conference — ICLR 2022 Submitted_

### Official Review · Reviewer_8DqH · 2021-10-27

**Correctness:** 2
**Technical Novelty And Significance:** 2
**Empirical Novelty And Significance:** 3
**Recommendation:** 6
**Confidence:** 4

**Main Review:**

Strengths
1. The authors conduct a lot of experiments to validate that simply calibrating the batch statistics at test-time can improve the model performance in different settings, such as robustness to corruptions and domain generalization.
2. The analysis in Figure 1 is inspiring and interesting.

Weaknesses
1. Lack of technical novelty.
- The idea of mixing source and target statistics has been proposed by Schneider et al. (2020). I don’t see how the so-called alpha-BN differs from it.
- It seems that the class correlation minimization loss is directly taken from Jin et al. (2020). I think the authors should not claim to use it as a contribution.
2. Method
- The logic flow from Section 4 to Section 5 is a bit weird to me. We can know replacing source statistics with target statistics can indeed reduce covariate shift if we use that representation to train a classifier (Figure 1(b)), but simply replacing the statistics will hurt the performance (Table 2). Do these two results intuitively conclude that we should mix the statistics from both source and target domains? I don’t really think so. Also, the selection of the alpha values seems so ad-hoc, without any theoretical analysis. What’s more, the alpha value of the semantic segmentation task is kind of overfitting on the test set (Figure 3). Isn’t this cheating?
- While in the classification problem we know that directly replacing batch norm statistics with target statistics is bad (Table 2), it actually works quite well in semantic segmentation (Figure 3(b)). I wonder why. I think we need more analysis to understand this phenomenon.
- Can the class correlation minimization loss be applied to the semantic segmentation task as well?
3. Some claims are wrong
- In Table 1, not all the domain generalization methods support online adjustment (actually, only one recent method supports this). I would suggest adding another term called test-time batch statistics update (or something similar) for better clarity.
- In Section 4, the authors mention test-time normalization. But in the related work section, the authors actually use the same term to refer to both Schneider et al. (2020) and Nado et al. (2020). However, these two works apply test-time normalization differently. The former one does a very similar thing as what this paper does, while the latter one replaces batch norm statistics with target batch statistics. It is confusing to use a single term to refer to two different methods.
4. Some details are missing
- In Section 4 “Error of ideal target hypothesis”, the author claims that “we train a new classifier over the target representations with corresponding labels”. Please define “target representations” here.
- In Section 7 “Discovering a better pre-trained model”, how to use alpha-BN to get a better pre-trained model for fine-tuning remains unclear to me.
5. Other comments
- The ERM baseline on semantic segmentation is too bad. Currently, GTA5 -> Cityscapes only gives 29% mIoU by using a ResNet-50 DeepLabv3. However, I know that we can easily get a 35~37% mIoU by using a ResNet-50 DeepLabv2, which is actually a weaker model. I think there must be something wrong with either the model setting or training schedule. Showing great improvement over a bad baseline is not that meaningful.



**Summary Of The Paper:**

This paper presents a method to calibrate batch normalization statistics at test time to improve a model’s cross-domain generalization ability under covariate shifts. The authors validate the effectiveness of the method on several datasets and tasks. The authors also conduct some interesting analyses to help better understand the method.

**Summary Of The Review:**

- The main concern is the lack of technical novelty. I would say both of the two main components in this method have been proposed by others previously. In this case, this paper over-claims its contributions. It will be better to reshape the paper as an empirical finding paper instead of a method-proposing paper.
- Also, semantic segmentation is so different from image classification, so that I am not sure if the conclusions we get from image classification can safely transfer to semantic segmentation. For example, the mismatch between Table 2 and Figure 3(b), as I mentioned in the weakness section as well. And it is also quite weird that another important component, class correlation minimization, is only applied on classification but not segmentation. So I would suggest the authors either discard the semantic segmentation and fully focus on image classification, or conduct additional experiments and analysis on semantic segmentation to have a deeper understanding.

---

> ### Author Response · Authors · 2021-11-12
> **Response to Reviewer #8DqH (Part 1)**
>
> Thanks for your positive feedback on our insight into the failures of T-BN. We address the issue below.
>
> Q1: Lack of technical novelty.
>
> A1.1: Firstly, we share similar formulation with (Schneider et al., 2020) and we have already mentioned it in Section 5: “Similar formulation is also proposed by Schneider et al. (2020) for alleviating the estimation error caused by small batch size. However, we discover that even with a large batch size (e.g., 200), T-BN also yields inferior performance on the large distribution shift (e.g., Office-Home)”. The key difference is that we have totally different motivations. Schneider et al. (2020) mainly concerns the estimated error caused by small target batch size, and validate it on image corruption benchmarks, where the corruptions are algorithmically generated. However, we mainly focus on the nature covariate shift from real-world scenarios. If the assumption on (Schneider et al., 2020) are valid, T-BN could perform well with a large target batch size. However, in this paper, we show that even with a large target batch size, T-BN still gains a inferior performance on the real-world distribution shift scenarios (e.g., VLCS, PACS and Office-Home), which motivates us to elaborate why T-BN fails in Section 4. We think the main contribution of this paper is providing a primary investigation on why T-BN fails on the real-world distribution shift scenarios, and show Alpha-BN consistently improves over a wide range of datasets, while previous methods ((Schneider et al. (2020) & TENT) only validate their methods on image corruptions, synthetic-to-real benchmarks and digits.
>
> A1.2: Secondly, the CORE loss is taken from (Jin et al., 2020) and we have stated it in the manuscript. We did not say that this loss is proposed by us, but we show that equipped with Alpha-BN, CORE loss outperforms Entropy loss adopted by TENT. In the revised manuscript, we revise the statement on the contribution part, and further provide a theoretical insight on why CORE loss outperforms Entropy loss and thorough ablation study in **Appendix C**.
>
> Q2.1: (1) The insight for Alpha-BN (2) The selection of alpha. (3) About Figure 3.
>
> A2.1: (1) Let us denote conventional inference as Alpha-BN($\alpha=1$) since it fixes the source statistics, and T-BN as Alpha-BN($\alpha=0$). $\alpha=1$ indicates that there exists no mismatch between batch statistics and model parameters, but the model will suffer from the distribution shift. $\alpha=0$ indicates that the distribution shift is alleviated but the batch statistics mismatch the model parameters, thus hurting the discriminative structures. Therefore, there exists a trade-off between distribution shift alleviation and discriminative structures. This finding mentioned in Section 4 motivates us to present Alpha-BN in Section 5. We think mixing the source and target statistics is easy to understand. We are happy to discuss with you if you have any other methods.
>
> (2) About the selection of the alpha values is discussed in **Section 7 (the third discussion)**. We select $\alpha$ by grid search (from 0 to 1 with an interval of 0.1). It may sound disappointing but we find the value of $\alpha$ is robust to task. To evaluate this statement, we conduct comprehensive experiments on both classification and segmentation, and observe that $\alpha=0.9$ for classification and $\alpha=0.7$ for segmentation is a good choice. From figure 3(b), we can find $\alpha=0.7$ is not the optimal but a good choice for all tasks. Therefore, the reported results are obtained by the fixed value of $\alpha$ rather than the task-tuned value of $\alpha$. Developing the learnable $\alpha$-BN with an automatically adjusted $\alpha$ is inspiring but difficult, thus we leave it for future work.
>
> (3) For the third question ” What’s more, the alpha value of the semantic segmentation task is kind of overfitting on the test set (Figure 3). Isn’t this cheating?”, we want to clarify that it is not called overfitting. Figure 3 (b & c) shows that the performance rises and then falls as the alpha value decreases from 1 to 0, which is consistent with our key insight. The performance rises first indicates that involving target statistics helps alleviating the distribution shift. The drop in performance afterwards suggests that completely discard the source statistics hurts the discriminative structures. This phenomenon is not called overfitting, but the key insight of our paper.

---

> ### Author Response · Authors · 2021-11-12
> **Response to Reviewer #8DqH (Part 2)**
>
> Q2.2: Confusion about Table 2 and Figure 3(b)
>
> A2.2: We are sorry about this confusion. In fact the two are not contradictory. As previous methods T-BN and TENT have demonstrated T-BN works well on two scenarios: image corruptions (the corruptions are algorithmically generated) and synthetic-to-real adaptation (the source domain data are artificially generated). That is why T-BN seems work well in Figure 3(b), since the task in Figure 3(b) belongs to the synthetic-to-real adaptation. However, previous works ignored the most important scenario: the nature distribution shift between real-world datasets. One of the major contributions of our paper is to show T-BN fails in this scenario and give a primary explanation as to why it fails (Section 4). And this finding actually motivates Alpha-BN. The experimental results in Table 2 are based on the nature distribution shift between real-world datasets, while the experimental results in Table 2 are based on synthetic-to-real adaptation. **In the revised version, we have included more discussions on this phenomenon in Section 7 (the second discussion).** Please check the revised manuscript for more details. Thanks a lot!
>
> Q2.3: Can CORE loss be applied to segmentation task as well?
>
> A2.3: Thanks for your suggestions! **We have added corresponding experiments in Appendix D.3 and Table 9**. Just as Reviewer #cBnx said, “ If I correctly understand it, CORE can not be used in semantic segmentation (as with the TENT can not)”. Both CORE and TENT shows limited improvement on semantic segmentation, but CORE seems like a little better than TENT. Recent advances in domain adaptive semantic segmentation also revealed that the universal unsupervised loss is not suitable for semantic segmentation, since this task has many intrinsic challenges like class-imbalance, boundary confusion and so on. Recently, we find a well-designed self-training test time adaptation method is effective for semantic segmentation. To avoid overly redundant articles, we leave these findings on future works.
>
> Q3: Some claims are wrong.
>
> A3: Thanks for your suggestions! We adopt the suggestions provided by Reviewer #cBnx, classify Alpha-BN into optimization-free test time adaptation to avoid misunderstanding. As for the confusion of references, we now use (Nado et al., 2020) to represent T-BN. Thanks again!
>
> Q4: Some details are missing.
>
> A4: (1) “target representations” means the representations are obtained by target domain data. **We have added the explanation in the revised version.** (2) "Discovering a better pre-trained model" is a wrong claim. **We have changed it to "$\alpha$-BN gains more transferable representations" in the revised version**.
>
> Q5: The ERM baseline on semantic segmentation is weak.
>
> A5: Firstly, we use the ERM baseline provided by recent SOTA DG semantic segmentation methods RobustNet. Secondly, to the best of our knowledge, using ResNet-101 and DeepLabV2 can get a ~35% mIoU rather than ResNet-50 and DeepLabV2. For instance, ERM model with 34.2 mIoU on (Kang et al., 2020), 33.8 mIoU on (Teja S et al., 2021) and 36.6 mIoU on (Zheng et al., 2021) are reported. All of them use ResNet-101 rather than ResNet-50. Our backbone is weaker than ResNet-101+DeepLabV2, so a lower ERM baseline performance is deserved. If you can provide the corresponding reference (ResNet50+DeepLabV2), we are happy to add the experiment to validate our Alpha-BN. In the revised manuscript, **we have included the result of GTA5->Cityscapes with ResNet101+DeepLabV2 backbone in Appendix D.2.**
>
> Thanks again for your precise suggestions and comments! Please let us know if there are further points we can discuss!
>
> Reference:
>
> (Kang et al., 2020) Kang et al. "Pixel-Level Cycle Association: A New Perspective for Domain Adaptive Semantic Segmentation." Advances in Neural Information Processing Systems 33 (2020).
>
> (Fleuret el al., 2021) Fleuret et al. "Uncertainty Reduction for Model Adaptation in Semantic Segmentation." Proceedings of the IEEE/CVF Conference on Computer Vision and Pattern Recognition. 2021.
>
> (Zheng et al., 2021) Zheng, Zhedong, and Yi Yang. "Rectifying pseudo label learning via uncertainty estimation for domain adaptive semantic segmentation." International Journal of Computer Vision 129.4 (2021): 1106-1120.

---

> > ### Comment · Reviewer_8DqH · 2021-11-16
> > **Highlight the changes**
> >
> > Thanks for your reply! I wonder if you can use a different color (e.g., red or blue) to highlight the changes you made in the revision.

---

> > > ### Author Response · Authors · 2021-11-17
> > > **We have highlighted the revisions in the new version.**
> > >
> > > Thanks for your suggestion!  The changes in new version are highlighted in red underlining.

---

> > ### Comment · Reviewer_8DqH · 2021-12-01
> > **Response**
> >
> > Thank you for your response and revision. I read all reviews, responses, and the revised paper. I think this paper indeed provides some interesting analysis and thorough experiments. Although the technical contribution/novelty is a bit limited, I think this paper has some empirical contributions. Thus, I am willing to increase the score.

---

### Official Review · Reviewer_H2Ln · 2021-11-01

**Correctness:** 3
**Technical Novelty And Significance:** 3
**Empirical Novelty And Significance:** 3
**Recommendation:** 5
**Confidence:** 4

**Main Review:**

Strengths:
- Proposed method can be easily applied on top of any base model for performing test-time adaptation
- Strong empirical results across a range of datasets and tasks, especially image segementation, outperforming recent methods
- Some insight into the failures of T-BN is described

Weaknesses:
- It is unclear how the parameter $\alpha$ should be set for new datasets. The chosen values happen to achieve the best performance according to Figure 3 (for segmentation), but how was this chosen and how should these be chosen in practice for a new dataset when the test performance is unknown? How $\alpha=0.9$ was selected for classification tasks was also not described in the text.
- A major contribution in the paper is the $\alpha$-BN method of combining source and target batch statistics, which the paper argues improves upon T-BN. However, the precise effect of this procedure versus the orthogonal class correlation optimization (CCO) was not examined in the experiments. In particular, each of the $\alpha$-BN and CCO components can be incorporated/compared with the previous TENT method that uses T-BN and entropy. An ablation study swapping out each respective component will show the significance of each part (e.g. T-BN + CCO, $\alpha$-BN + Entropy).
- There appear to be some inconsistencies with the experimental evaluation as detailed below:
  * For segmentation experiments, it appears not all of the methods share the same backbone model, which has a large impact on performance, e.g. SW (Pan et al. 2019) reports using "DeepLab-v2 [2] model with VGG16 [25] backbone", but this paper uses DeepLabV3 with ResNet-50 as stated on page 8. Moreover, the result for SW in the GTA5 -> Cityscapes setting is 29.9 in Table 7, which differs from the result reported in the paper (Table 5; AdaptSetNet-SW - mIOU 35.7). These differences should be clarified during the discussion period. It should also be clarified whether the same backbone is used for comparisons in all other experiments as well.
  * TENT is included in other comparisons but not in the segmentation results - is there some particular reason why? The original TENT was also not included in Table 6 - how does it perform?
- Comparison to recent DG for segmentation baseline is missing - FSDR: Frequency Space Domain Randomization for Domain Generalization, CVPR 2021.
- Presentation wise, some parts of the text should either be supported by evidence or toned-down in terms of the claims, e.g. "guaranteed discriminative representations" (this suggests there will be a proof), "suppresses the confident false predictions ... providing a more robust optimization" (some empirical evidence of this should be provided). Also, Section 4 seems a bit tangential to the rest of the paper as it does not directly motivate $\alpha$-BN. Does the proposed $\alpha$-BN method perform better than T-BN in evaluations provided in Section 4? This will provide better motivation for the $\alpha$-BN method and improve the flow of the paper.

Other comments/suggestions:
- The justification for comparing the proposed method with other DG methods is that Pandey et al 2021 does some optimization at test time. However, they do not modify the model parameters unlike the proposed method and other test-time adaptation approaches. Moreover, the method of Pandey et al 2021 was not included in the comparisons.
- It would be useful to provide some insight into why the proposed method does well in some cases compared to others.
- Statistical tests for significance should probably be done to compare all methods rather than just the baseline, and on all tasks.

**Summary Of The Paper:**

This paper presents a domain adaptation algorithm for the recently introduced test-time adaptation setting by adapting batch-norm parameters. The algorithm has two key components: first, it adapts batch-norm statistics using a linear combination of source and estimated target domain statistics; second, it uses a class correlation optimzation loss to optimize batch-norm parameters on the test batches. These design choices differ from previously described methods. The resulting algorithm is evaluated on a range of datasets and tasks including common corruptions and the DomainBed benchmark for image classification, as well as a benchmark for image segmentation.




**Summary Of The Review:**

Overall this paper proposes a simple and effective method for test-time adaptation that seems to outperform the previously described TENT method. However, key questions about how the $\alpha$ parameter should be set in practice and issues with the experimental evaluations should be resolved before the paper is ready for publication. A more thorough examination of the effect of $\alpha$-BN and CCO components will also provide more confidence in the contribution.

**Post response update:** I have read the other reviews and responses from the authors. The additional ablation studies address my concerns about the effect of the individual components, but as pointed out by the other reviewers as well, technical novelty is somewhat limited as both components have existed in the literature although they are not applied to this particular setting. Also, it is unclear how the $\alpha$ parameter can be set in practice. Thus, while this work presents an interesting empirical study, I think it is borderline and I lean towards rejection.

---

> ### Author Response · Authors · 2021-11-12
> **Response to Reviewer #H2Ln (Part 1)**
>
> Thanks for your time and thoughtful comments! We address your concerns as follows:
>
> Q1: About the selection of hyper-parameter $\alpha$.
>
> A1: Thanks for your comments! **We have included some discussion and more empirical results in Section 7 and Appendix D.4**. We select $\alpha$ by grid search (from 0 to 1 with an interval of 0.1). It may sound disappointing but we find the value of $\alpha$ is robust to task. To evaluate this statement, we conduct comprehensive experiments on both classification and segmentation, and observe that $\alpha=0.9$ for classification and $\alpha=0.7$ for segmentation is a good choice. From figure 3(b), we can find $\alpha=0.7$ is not the optimal but a good choice for all tasks. Therefore, the reported results are obtained by the fixed value of $\alpha$ rather than the task-tuned value of $\alpha$. Developing the learnable $\alpha$-BN with an automatically adjusted $\alpha$ is inspiring but difficult, thus we leave it for future work.
>
> Q2: About the ablation study on CORE and TENT. (e.g., T-BN+COO and Alpha-BN+Entropy)
>
> A2: Thanks for your suggestions! **We have added corresponding ablation study on Appendix D.1**. We observe that both T-BN+Entropy and T-BN+COO are worse than ERM. Equipped with our Alpha-BN, COO outperforms Entropy. Just as Reviewer #A9FS said, Alpha-BN provides a better initialization than T-BN. We highlight that a better initialization is really important since the supervision signal comes from the predictions. The better initial predictions result in better supervision signals on the unlabeled target samples. That is why both COO and Entropy performs badly with T-BN.
>
> Q3: About the inconsistence performance of SW.
>
> A3: For semantic segmentation, we cite the performance of compared methods from the latest state-of-the-art method RobustNet (Choi et al., 2021), which is also named ISW. RobustNet re-implemented SW with ResNet50 + DeepLabV3+, and we have also run their released code and find the performance is consistent with the reported ones. Since the backbone is different, the corresponding results should rightly be different. The reported results in Table 7 are obtained by ResNet50 + DeepLabV3+ backbone.
>
> Q4: Whether the same backbone is used for comparisons in all other experiments?
>
> A4: Absolutely yes. The used backbones have already shown in Table 3.
>
> Q5: How about original TENT’s performance in classification and segmentation.
>
> A5: Thanks for your suggestions! **We have included the classification results of TENT in Appendix D.1**. Since TENT is based on T-BN, which shows inferior performance compared to Alpha-BN, CORE significantly outperforms the original TENT. **For semantic segmentation, the corresponding results are shown in Appendix D.3**.
>
> Q6: A missing reference: FSDR
>
> A6: Thanks for your suggestions! Since we adopt RobustNet (Choi et al., 2021) to evaluate Alpha-BN, whose backbone is different from FSDR’s, we did not compare with FSDR in the submitted manuscript. **In the revised version, we have included the comparison with FSDR and other methods with backbone DeepLabV2+ResNet101 on task GTA5->Cityscapes. Please refer to Appendix D.2 for more details.**
>
> Q7: About some too casual claims.
>
> A7: Thanks for your suggestions! (1) we have revised the claim “guaranteed ...”. (2) The empirical results are provided in the third paragraph of Section 6.4. CORE loss significantly outperforms Entropy loss on the most challenging benchmark DomainNet. (3) The results of Alpha-BN on VLCS, PACS and Office-Home are shown in Table 6, which significantly outperform T-BN (9.5% averaged accuracy improvement). We provide a primary investigation on T-BN and the failures of T-BN actually motivates our Alpha-BN. We are sorry that our writing confused you, we have revised this section on the revised manuscript.

---

> > ### Author Response · Authors · 2021-11-12
> > **Response to Reviewer #H2Ln (Part 2)**
> >
> > Q8: About the method of (Pandey et al, 2021)?
> >
> > A8: We want to clarify that we did not say (Pandey et al, 2021) does some optimization at test time. What we say is that (Pandey et al, 2021) performs online adjustment rather than online optimization (Table 1). We are sorry that “online adjustment” is a confusing term. As suggested by Reviewer #cBnx, we use “optimization-free test time adaptation” to classify (Pandey et al, 2021) and the proposed Alpha-BN, and “optimization-based test time adaptation ” to classify TENT and CORE. Therefore, we compare Alpha-BN with other DG methods, and compare CORE with TENT only in Table 6. The comparison is fair. Another concern is that why we did not compare Alpha-BN with (Pandey et al, 2021). This is because the implementation and model selection have significant effect on the performance of DG, and the recent work DomainBed (Gulrajani & Lopez-Paz, 2020) shows simplest ERM outperforms most DG methods. After that, researchers usually adopt DomainBed to evaluate the proposed method, and so do we. However, (Pandey et al, 2021) is not implemented on DomainBed, so the comparison with (Pandey et al, 2021) is unfair. For instance, the PACS result of RSC (Huang et al., 2020) on (Pandey et al, 2021) is 87.8%, while it only obtains 85.2% on DomainBed.
> >
> > Q9: Provide some insight into why the proposed method does well in some cases compared to others
> >
> > A9: Thanks for your suggestions! We have added discussions on why Alpha-BN outperforms T-BN and CORE loss outperforms Entropy loss. **Please refer to Section 7 (the first discussion) for more details.**
> >
> > Q10: Statistical test on all tasks and all compared methods.
> >
> > A10: Thanks for your suggestions! We have included the significance test results in the revised version, we use ‘*’ to represent that Alpha-BN is statistical significant (**see Table 4 & 5 & 6 and the full results in Appendix B**). Please check the revised manuscript.
> >
> > Thanks again for your precise suggestions and comments! Please let us know if there are further points we can discuss!

---

> ### Comment · Reviewer_H2Ln · 2021-11-25
> **Response to Revision**
>
> The revision and response has addressed my concerns about evaluation and the contributions of each component. The ablation studies are thorough and the clarification on backbones and performance are appreciated.
>
> My main remaining concern is regarding how the parameter $\alpha$ should be set. It is mentioned in the response that this was set by grid search, but what is the evaluation criteria used here? It appears to me that it is the *test error*, as alluded to by reviewer 8DqH as well, which is not quite right. Further, the response says that $\alpha$ is robust to task, but the plots in Figure 5 show potentially substantial variation in performance (5-10%), even more sometimes over the middle range of $\alpha$, which is comparable if not more than the margin of improvement over TENT.

---

> > ### Author Response · Authors · 2021-11-26
> > **We determine the value of $\alpha$ by one task, and fix it for the remaining tasks. We did not tune the value of $\alpha$ for each task.**
> >
> > Thanks for your feedbacks! We are encouraged that our response has addressed many of your concerns. We elaborate the evaluation criteria and clarify the claim "$\alpha$ is robust to task" as follows:
> >
> > We are sorry that we did not mention the evaluation criteria in the paper. Firstly, in Section 3, we reveal that both $\alpha=0$ and $\alpha=1$ are not great choices. Therefore, we propose to balance the source and target statistics. For classification, we use the task $\mathcal{R} \rightarrow C$ in PACS dataset, which is also used to visualize the learned representation (Figure 1), to determine the value of $\alpha$. For this task, we determine the value of $\alpha$ by grid search, and find $\alpha=0.9$ is the best choice. Then, we set $\alpha=0.9$ for the other tasks. In short, we determine the value of $\alpha$ by grid search on one task, then fix it for the remaining tasks. For semantic segmentation, we use the task GTA5 $\rightarrow$ Cityscapes to determine the value of $\alpha$. It is noticing that the determined $\alpha$ is not the necessarily optimal choice for each task. For instance, $\alpha=0.5$ is better than $\alpha=0.9$ for task $\mathcal{R} \rightarrow A$ in PACS (Figure 5). We did not tune the value of $\alpha$ for different tasks to get the best performance since the target domain is unseen, but fix it for all tasks. The reported results are obtained by the fixed $\alpha$ rather than the task-tuned $\alpha$.
> > And we find that $\alpha=0.9$ and $\alpha=0.7$ are the acceptable choices for classification task and semantic segmentation task, respectively. We summerize this phenomenon as "$\alpha$ is robust to task".
> >
> > The substantial variation in performance in Figure 5 is the key insight and motivation of our method. $\alpha=1$ equals to the traditional vanilla inference setting, while $\alpha=0$ equals to the recently proposed T-BN. The clear degradation made by T-BN ($\alpha=0$) motivates the proposed $\alpha$-BN. The Figure 5 is consistent with our claims. From Figure 5, we can also observe that $\alpha=0.9$ is the great choice for all tasks, but not the necessarily optimal choice for each task.

---

### Official Review · Reviewer_A9FS · 2021-11-02

**Correctness:** 3
**Technical Novelty And Significance:** 2
**Empirical Novelty And Significance:** 2
**Recommendation:** 5
**Confidence:** 4

**Main Review:**

Strength:

The paper uses the same setting of the test-time BN methods to perform test time adaptation on unlabeled target mini-batch data. The method has been well-motivated by pointing out the limitations in SOTA methods, i.e., BN adaptation alleviates the domain statistics shift, but perturbs the discriminative structures. The paper provides comprehensive experiment results.

weakness:
 - They present the $\alpha$-blending formulation to mix up the source and target statics instead of directly using the target statistics. They empirically set the weight $\alpha$ =0.9 for image classification and 0.7 for image segmentation tasks. There is no discussion on how to select $\alpha$.

 - The paper proposes to update beta and gamma on the mini-batch with the Core loss. The main concern is that the proposed Core loss in EQ 5,  $L_{core}=  1 - \sum_j p_j p_j$,  is actually very similar to the entropy loss $L_{ent} = - \sum_j p_j log(p_j)$. The experimental results in the papers show the performance is better. However, there is no theoretical proof or explanation that the Core loss can necessarily get better results than the entropy loss. And, it has the same problem as the entropy loss: the proposed loss cannot prevent trivial collapsed solutions where all unlabeled samples are assigned the same one-hot encoding (which is the actual minima).

**Summary Of The Paper:**

The paper proposes a test-time batch normalization method for domain adaptation. They present a formulation $\alpha$-BN to calibrate the batch statistics by mixing up the source and target statistics with a fixed hyper-parameter of alpha. And, they propose to use the Core loss [ Jin et al. (2020)] to optimize the affine parameters of beta and gamma in BN layers instead of the entropy minimization used in the Tent method. The paper provides comprehensive experiment results.

**Summary Of The Review:**

The proposed method can be considered as an extension of Tent. Essentially, the optimal BN parameters ( $\mu$, $\sigma$, $\beta$, and $\gamma$) can be obtained by optimizing only the BN affine parameters ( $\beta$, and $\gamma$). The proposed alpha-BN actually provides a better initialization of the BN parameters for DA, which is why it can obtain better performance than T-BN. But, there is no proof that the proposed $L_{core}$ can necessarily achieve better results than Tent. I consider the proposed method is incremental. The authors should provide more explanation & discussion of the $L_{core}$ in the rebuttal.

---

> ### Author Response · Authors · 2021-11-12
> **Response to Reviewer #AF9S**
>
> Thanks for your precise suggestions and reviews. Here are our answers to your concerns.
>
> Q1: About the selection of hyper-parameter $\alpha$.
>
> A1: Thanks for your comments! **We have included some discussion and more empirical results in Section 7 and Appendix D.4**. We select $\alpha$ by grid search (from 0 to 1 with an interval of 0.1). It may sound disappointing but we find the value of $\alpha$ is robust to task. To evaluate this statement, we conduct comprehensive experiments on both classification and segmentation, and observe that $\alpha=0.9$ for classification and $\alpha=0.7$ for segmentation is a good choice. From figure 3(b), we can find $\alpha=0.7$ is not the optimal but a good choice for all tasks. Therefore, the reported results are obtained by the fixed value of $\alpha$ rather than the task-tuned value of $\alpha$. Developing the learnable $\alpha$-BN with an automatically adjusted $\alpha$ is inspiring but difficult, thus we leave it for future work.
>
> Q2: Theoretical insight on why CORE loss is better than Entropy loss, and how to prevent the trivial collapsed solutions.
>
> A2: Thanks for your suggestions! **We have included the theoretical insight on Appendix C**. The key insight is that CORE loss has much smaller gradient on the easy samples compared to Entropy loss. In other words, CORE loss prevents the easy samples dominate the optimization procedure. Another concern is that both CORE loss and Entropy loss may suffer from the trivial collapsed solutions. Firstly, we want to clarify that all unlabeled samples are assigned the same one-hot encoding is not the only actual minima. When all unlabeled samples are assigned the true labels (one-hot) is also the minima for CORE loss and Entropy loss. CORE loss is used to reduce the ambiguity of the target predictions and increases the class separation. We think what you what to ask is that “Will there be some samples that CORE loss provides wrong supervision signals?” This situation is inevitable for unsupervised methods like entropy minimization, pseudo-labeling and so on. But generally speaking, CORE loss consistently improves the performance on a wide range of benchmarks, and outperforms Entropy loss.
>
> Thanks again for your insightful reviews. $\alpha$-BN provides a better initialization of BN parameters is a great view to understand it. A better initialization is really important for unsupervised online learning since the supervision signals usually come from the predictions (e.g., CORE, TENT, pseudo-labeling). A better initialization provides a better predictions, thus giving a better supervision signals for further adaptation. Please let us know if there are further points we can discuss!

---

> > ### Author Response · Authors · 2021-11-27
> > **We really hope to have a further discussion with you to see if our responses address your concerns.**
> >
> > Dear Reviewer #AF9S:
> >
> > Thanks for your valuable and insightful reviews! We have revised our manuscript based on your suggestions. Given the ICLR final discussion deadline (11/29) is approaching, we really hope to have a further discussion with you to see if our responses address your concerns. Thank you!

---

### Official Review · Reviewer_cBnx · 2021-11-03

**Correctness:** 3
**Technical Novelty And Significance:** 3
**Empirical Novelty And Significance:** 3
**Recommendation:** 6
**Confidence:** 5

**Main Review:**

**Strengths**

1. Both proposed modifications are technically sounds.

2. Empirical evaluation on many experimental settings show reliable (yet not revolutionary) performance improvement by the proposed methods.

3. This paper is well written and easy to follow.


**Weaknesses**

1. The proposed method is technically sound but not a revolutionary idea. Regarding CORE, difference from (Jin et al., 2020) is not clear from the manuscript.

2. Table 1 is confusing, since DG usually assumes the existence of source data, and does not adapt online usually. Similarly, protocol DG in Table 6 is a little all methods listed in the first block of Table 6 does not use any test-time adaptation. I think it is better to call it optimization-free test-time adaptation to clarify the difference with optimization-based TTA like TENT.

3. From my personal experience, I am a little bit  surprised that simple T-BN significantly worsened the performance in Table 2. In my past implementation, I also observed slight performance degradation on VLCS (say 3%) yet slight performance improvement on PACS (say 1.5%) by T-BN.
What is the batch size in the experiment? Is there any class imbalance among test bach in your evaluation? Any other detail that could change the results?

4. Table 6 shows that using Tent in DomainNet worsens the performance yet CORE stable increases the performance. They claim that this is because the domain gap in DomainNet is extremely large, which might be a true yet too casual statement not suitable for the academic paper. It should be justified qualitatively or quantitatively why Tent fails to handle the huge distribution gap, and DomainNet has a huge distribution gap.
Besides I am suspicious that the performance gain is caused by the hyperparameter selection. It should be better to include the hyperparameter sensitivity of Tent and CORE to fully show the CORE is indeed superior one.

5. Table 7 lacks comparison with T-BN, therefore it is not clear whether the performance gain caused by the modification or the nature of test-time normalization. Besides, If I correctly understand it, CORE can not be used in semantic segmentation (as with the Tent can not), which should be elaborated in the manuscripts.

A minor comment: In the last sentence in 6.4, I think Tent should be CORE.


**Summary Of The Paper:**

This paper presents several modifications over test-time normalization and test-time adaptation, which swap the statistics of the BN layer during test-time to handle the distributional difference. Specifically, they propose alpha-BN to balance the source statistics and target statistics, and CORE to further refine the transformation parameters of BN, which is similar to prior test-time adaptation methods.


**Summary Of The Review:**

Overall I think this is well-written and an acceptable paper with solid empirical results. I am happy to increase my score if the response clarifies my concerns or I misunderstood some points.

---

> ### Author Response · Authors · 2021-11-12
> **Response to Reviewer #cBnx**
>
> Thanks for your detailed comments and positive feedback. We are encouraged that you found our method to be technically sound. We address the specific questions one by one. The mistake in the last sentence in Sec. 6.4 has been revised.
>
> Q1: What’s the difference between CORE and (Jin et al., 2020)?
>
> A1: CORE contains $\alpha$-BN and CORE loss, the CORE loss is taken from (Jin et al., 2020), and we have mentioned it in the manuscript. In the revised version, we revise the the statement in Section 4 to avoid overclaiming our contributions, and further provide a theoretical insight on why CORE loss outperforms Entropy loss in **Appendix C**.
>
> Q2: Optimization-free test-time adaptation is a better protocol name for $\alpha$-BN.
>
> A2: Thanks for your comment! **We have updated it in the revised manuscript**. What we want to convey is that the comparison between $\alpha$-BN and other DG methods is fair since (1) $\alpha$-BN introduces very little additional inference time, (2) some recent DG works also adapt the model during inference. Except for (Pandey et al., 2021) that we have mentioned, another recent DG work (Du et al., 2021) fuses multi-source models during inference. We think optimization-free test time adaptation is an interesting direction for DG since it involves target domain information and hardly affects inference time.
>
> Q3: About the experimental settings of T-BN on common DG benchmarks like VLCS and PACS.
>
> A3: The settings of T-BN is the same as our Alpha-BN: batch size=64, and the target data are shuffled to prevent the class imbalance issue. The results on PACS are similar to yours (1.6% drop), but the results on VLCS are different. **We have included detailed results of T-BN on VLCS, PACS and Office-Home in Table 8**.
>
> Q4: (1)Too causal statement on DomainNet. (2) Is the superiority of CORE compared to TENT caused by the parameter selection?
>
> A4.1: Thanks for pointing out the too causal statement! We have modified it in the revised version. **We have also provided a theoretical insight on why CORE loss outperforms Entropy loss in Appendix C**.
>
> A4.2:The experimental settings for CORE and TENT are the same. Firstly, the results of TENT in Table 6 are based on $\alpha$-BN ($\alpha$-BN+Entropy loss), and we have included the other variants (the original TENT: T-BN($\alpha=0$)+Entropy loss) in Table 8 & 9. We can observe that CORE ($\alpha$-BN+CORE loss) significantly outperforms original TENT (T-BN+Entropy loss). Both equipped with $\alpha$-BN, CORE loss still outperforms Entropy loss. We think the added results imply that CORE is superior compared to TENT. As for the hyper-parameter, there is no hyper-parameter on both CORE loss and Entropy loss. Please let us know which parameter you would like to investigate further. We are glad to add corresponding studies.
>
> Q5: The semantic segmentation results.
>
> A5: Thanks for your comments. **We have added the detailed results of T-BN in Table 7**. Alpha-BN outperforms T-BN, but the improvement on semantic segmentation seems limited compared to classification. However, the results in Table 7 are obtained from synthetic-to-real adaptation scenario. We recently find Alpha-BN outperforms T-BN on the real-to-real adaptation scenario (e.g., Cityscapes -> {BDD-100K, Mapillary}), **which is also added in the revised manuscript (see Table 7).** This finding is consistent with the classification ones: T-BN works well on VisDA-2017 (synthetic-to-real) but fails on VLCS, PACS, Office-Home and so on. As for the optimization-based test-time adaptation on semantic segmentation, both TENT and CORE brings limited improvements. (**We have included the results in Appendix D.3 and Table 9**) For semantic segmentation, we recently find that a well-designed self-training approach equipped with Alpha-BN performs well on both synthetic-to-real and real-to-real adaptation scenarios under test-time adaptation setting. To avoid overly redundant articles, we leave these findings on future works.
>
> Thanks again for your valuable comments! You are really familiar with TTA, and we are really happy to see your precise suggestions. Please let us know if there are further points we can discuss!
>
> Reference:
>
> (Du et al., 2021) Du, Zhekai, et al. "Learning Transferrable and Interpretable Representations for Domain Generalization." Proceedings of the 29th ACM International Conference on Multimedia. 2021.

---

> > ### Author Response · Authors · 2021-11-27
> > **We really hope to have a further discussion with you to see if our responses address your concerns.**
> >
> > Dear Reviewer #cBnx:
> >
> > Thanks for your valuable and insightful reviews! We have revised our manuscript based on your suggestions. Given the ICLR final discussion deadline (11/29) is approaching, we really hope to have a further discussion with you to see if our responses address your concerns. Thank you!

---

### Author Response · Authors · 2021-11-12
**Response to All Reviewers**

Thanks for your valuable reviews! We are encouraged by your positive feedbacks. We have uploaded the revised manuscript. The revisions are highlighted in red underlining. Due to time constraints, we have only added the requested content. We are still working on organizing our paper for future readers!

The main revisions are listed as follows:

(1) The discussion on "how to choose the value of $\alpha$" for a new dataset.

(2) A theoretical insight on why CORE loss outperforms Entropy loss.

(3) More ablation study between CORE and TENT.

(4) The results of CORE and TENT on semantic segmentation.

(5) The discussion about which scenarios T-BN is and is not good at.

Here we want to restate our key contributions.

We are the first work to evaluate T-BN on the nature distribution shift of real-world datasets (e.g., VLCS, PACS and Office-Home). And we provide a primary investigate into the failures of T-BN: a trade-off between covariate shift alleviation and discriminative structure preservation. Based on this finding, we present $\alpha$-BN to perform test-time batch statistics calibration, and consistently improves the performance of common DG models. Furthermore, we find the class correlation optimization loss (Jin et al., 2020) performs better than the entropy loss adopted by previous SOTA method TENT on a wide range of benchmarks, and provide a theoretical insight on it.

---

### Decision · Program_Chairs · 2022-01-20

**Decision:**

Reject

**Comment:**

The paper builds on ideas in test-time adaptation and test-time normalization to improve performance under covariate shift. Concretely, the paper proposes (i) alpha-BN, a method to calibrate batch statistics by mixing source and target statistics and (ii) test-time adaptation using the CORE loss (which was proposed by Jin et al., 2020). The authors compare the the proposed approach to existing approaches on multiple benchmarks.

The reviewers found the idea interesting and appreciated the additional ablations. The main concerns were around novelty (as the idea is closely related to prior work in test-time adaptation and normalization) and hyperparameter selection (e.g. how to choose alpha in practice).  Overall, the reviewers and I felt that the current version falls slightly below the acceptance threshold. I encourage the authors to revise and resubmit to another venue.

Minor comment about Appendix C (this didn't affect the score, just a suggestion for future revisions):
I think it might be interesting to include other alternatives to cross-entropy that downweight easy examples, cf. focal loss https://arxiv.org/abs/1708.02002 and https://arxiv.org/abs/2002.09437. I'm curious to see if CORE and focal loss consistently outperform cross entropy.